# Non-differentiable Regularization for Heavy-tailed Differentially Private Stochastic Convex Optimization

## Abstract

Recently, differentially private stochastic convex optimization (DP-SCO) with heavy-tailed (HT) (sub)gradients has attracted increasing attention. Weaker than the uniform Lipschitz continuity assumption, the HT assumption on the stochastic (sub)gradients aligns more closely with real-world data distributions. However, few existing methods can handle the non-differentiable regularization (NDR) for HT DP-SCO. The main difficulty lies in achieving a low excess population loss and a low computational complexity simultaneously. In this work, we propose a novel forward-backward splitting approach to tackle NDR for HT DP-SCO, abbreviated by NDR-HT. It satisfies concentrated differential privacy, achieves an asymptotically optimal excess risk bound (up to logarithmic factors), and requires only $\mathcal{O}(n \log n)$ gradient evaluations, which is a lower computational complexity than those of existing state-of-the-art approaches. Furthermore, NDR-HT achieves linear convergence up to an additive approximation error and avoids solving complex subproblems in each iteration. Extensive experiments on both synthetic and real-world data show that our approach is effective and efficient in solving HT DP-SCO with NDR.

## 1 Introduction

The differentially private stochastic convex optimization (DP-SCO) problem is extensively investigated in machine learning (Chaudhuri & Monteleoni, 2008; Duchi et al., 2013; Bassily et al., 2014; Ullman, 2015; Cheu et al., 2021; Asi et al., 2021; Lowy & Razaviyayn, 2023; Asi et al., 2024). Given a data set $\mathcal{D} := \{s_1, s_2, \ldots, s_n\} \subseteq \mathcal{S}$ consisting of $n$ independent and identically distributed (i.i.d.) samples drawn from an unknown distribution $\mathcal{P}$ over a sample space $\mathcal{S}$, the objective of DP-SCO is to develop a differentially private algorithm that outputs an approximate solution $\hat{x}$ for the following optimization problem:

$$\min_{x \in \mathcal{X}} F_{\mathcal{P}}(x) := \mathbb{E}_{s \sim \mathcal{P}}[f(x; s)], \tag{1}$$

where $f : \mathbb{R}^d \times \mathcal{S} \to \mathbb{R}$ denotes a convex **fidelity** and $\mathcal{X} \subseteq \mathbb{R}^d$ is a convex and compact parameter domain. The quality of the solution $\hat{x}$ is evaluated by the following **excess population loss** (**EPL**, Feldman et al. 2020):

$$F_{\mathcal{P}}(\hat{x}) - F_{\mathcal{P}}(x^*), \tag{2}$$

where $x^* = \arg\min_{x \in \mathcal{X}} F_{\mathcal{P}}(x)$ is an optimal point of model (1). In this work, we focus on a regularized variant of model (1), which incorporates a **non-differentiable regularization (NDR)** $g(x)$ into the fidelity and leads to the following **loss function**:

$$\min_{x \in \mathcal{X}} \mathcal{F}(x) := \{F_{\mathcal{P}}(x) + g(x)\}. \tag{3}$$

We assume that the fidelity $f(\cdot; s)$ is strongly convex and smooth on $\mathbb{R}^d$, while the NDR $g : \mathbb{R}^d \to \mathbb{R}$ is convex and Lipschitz continuous but not necessarily differentiable. Due to the non-differentiability of $g$, the loss function $\mathcal{F}$ is also non-differentiable, making (3) a typical model for non-differentiable DP-SCO problems that cannot be directly solved by standard gradient-based

methods. Specifically, the NDR model (3) is highly relevant in modern robust and distributed machine learning. Non-differentiable penalties, such as $\ell_1$ (LASSO), Group LASSO, or Total Variation, as well as structured constraints from LLM parameter efficiency techniques, such as Parameter-Efficient Fine-Tuning (PEFT, Han et al. 2024) and Low-Rank Adaptation (LoRA, Hu et al. 2022), are essential for model sparsity, interpretability, and efficiency. Existing DP methods designed for differentiable objectives cannot handle these NDRs directly, creating a critical need for frameworks that guarantee privacy while addressing the NDR model.

On the other hand, most existing works on DP-SCO are built upon the **Lipschitz assumption** (see Assumption 2) (Bassily et al., 2019; Feldman et al., 2020; Asi et al., 2021; Bassily et al., 2021; Lowy & Razaviyayn, 2023). While this assumption is convenient for ensuring privacy guarantees and deriving bounds on the EPL, it may not hold in all the practical applications. Worse still, the Lipschitz constant $L_f$ could be excessively large or even unbounded due to outliers in the data (Crovella et al., 1998; Markovich, 2008; Woolson & Clarke, 2002). In such cases, the resulting EPL, which depends on $L_f$, could be either overestimated or entirely invalid. To address this problem, several works investigate DP-SCO in the weaker **heavy-tailed (HT) assumption** (see Assumption 1) (Wang et al., 2020; Kamath et al., 2022; Lowy & Razaviyayn, 2023; Asi et al., 2024), which reduces sensitivity to outliers. In this work, we assume that the distribution $\mathcal{P}$ is HT if not specified.

However, relaxing the Lipschitz assumption to the HT assumption also inspires new challenges for ensuring $\rho$ **concentrated differential privacy ($\rho$-CDP)**, such as high EPL, high computational complexity, and inability to handle NDR (see Table 1). For example, the localized noisy clipped subgradient method (LNCSM, Lowy & Razaviyayn 2023) requires the Lipschitz assumption to handle NDR, and cannot asymptotically achieve the theoretical lower found of EPL (Asi et al., 2024). Additionally, it relies on a localization framework (Feldman et al., 2020) and queries $\tilde{O}(n^2 L_f^2/G_2^2)$ subgradient evaluations (where $G_2$ is defined in Assumption 1), which is computationally expensive. Although the noisy clipped stochastic gradient descent (NCSGD, Lowy & Razaviyayn 2023) method runs in linear time for smooth and strongly convex $f$, it cannot handle NDR. The population-level localization scheme (PLLS, Asi et al. 2024) relaxes the $L$-smoothness (see Definition 4) of NCSGD to the weaker differentiability of $f$, but requires higher computational complexity than NCSGD and cannot handle NDR.

Table 1: Comparisons between different $\rho$-CDP algorithms under the 2-heavy-tailed setting (the most general heavy-tailed setting) for $\omega$-strongly convex loss function models. The theoretical lower bound of EPL for model (1) with strongly convex $f$ is $\Omega\left(d\left(\frac{1}{n} + \frac{\sqrt{d}}{n\sqrt{\rho}}\right)\right)$, regardless of whether $f$ is smooth or not (Lowy & Razaviyayn, 2023). PLLS, NCSGD, and NDR-HT asymptotically achieve this lower bound, while LNCSM cannot. The parameters $L_{\nabla F}$, $\sigma$, $G_2$ and $\bar{\kappa}$ are defined in (6), (7), Assumption 1 and Proposition 5, respectively.

| Method | Fidelity | Non-differentiable Regularization | Lipschitz Assumption | Excess Population Loss | Computational Complexity |
|---|---|---|---|---|---|
| PLLS | Differentiable | Unavailable | Non-required | $\mathcal{O}\left(\frac{\log(\frac{1}{\sigma})}{\omega n} + \left(\frac{\sqrt{d}\log^{\frac{3}{2}}(\frac{1}{\sigma})}{\omega n \sqrt{\rho}}\right)\right)$ | $\mathcal{O}\left(\max\left(n^2, \frac{n^3 \rho}{d}\right)\right)$ |
| NCSGD | $L$-smooth | Unavailable | Non-required | $\mathcal{O}\left(\frac{1}{\omega n} + \left(\frac{\sqrt{d(L_{\nabla F}/\omega)\log(n)}}{\omega n \sqrt{\rho}}\right)\right)$ | $\mathcal{O}(n)$ |
| LNCSM | Non-differentiable | Available | Required | $\mathcal{O}\left(\frac{G_2^2}{\omega}\left(\frac{1}{n} + 1\right)\right)$ | $\tilde{O}\left(\frac{n^2 L_f^2}{G_2^2}\right)$ |
| **NDR-HT (ours)** | $L$-smooth | Available | Non-required | $\mathcal{O}\left(\frac{1}{\omega n} + \frac{\sqrt{d\log\left(n\sqrt{\rho}\right)/(1-\bar{\kappa}^2)}}{n\sqrt{\rho}}\right)$ | $\tilde{\mathcal{O}}(n)$ |

To address the above challenges, we propose a novel forward-backward splitting approach (named **NDR-HT**) to tackle NDR for HT DP-SCO (model Eq. 3). It achieves $\rho$-CDP, runs in asymptotically linear time, and attains an asymptotically optimal EPL bound (up to logarithmic factors). Our main contributions are:

**1.** We characterize the solution set to model (3) as the fixed point set of a proximal gradient operator.

**2.** We transform the above proximal gradient operator into a contractive forward-backward splitting operator, then exploit this contractivity to suppress the EPL under the $k$-HT assumption with the presence of NDR.

**3.** By exploiting the above contractivity property, NDR-HT requires only $\mathcal{O}(n \log n)$ gradient computations to achieve an asymptotically optimal EPL bound. Furthermore, it achieves a linear convergence rate up to an additive approximation error, and ensures $\rho$-CDP. In contrast to other state-of-the-art methods in Table 1, NDR-HT is the only method that attains the optimal EPL rate for model (3) without sacrificing computational efficiency.

## 2 PRELIMINARIES AND RELATED WORKS

### 2.1 PRELIMINARIES

We now provide the formal definitions of some key concepts in this paper. Two data sets $\mathcal{D} := \{s_1, s_2, \ldots, s_n\}$ and $\mathcal{D}' := \{s'_1, s'_2, \ldots, s'_n\}$ of the same size are *neighboring* if they differ in at most one element, denoted by $\mathcal{D} \sim \mathcal{D}'$.

**Definition 1** (($\epsilon, \vartheta$)-Differential Privacy, Dwork et al. 2006). A randomized algorithm $\mathcal{A} : \mathcal{S} \to \mathcal{X}$ satisfies ($\epsilon, \vartheta$)-differential privacy (($\epsilon, \vartheta$)-DP with $\epsilon, \vartheta \geqslant 0$), if for all pairs of neighboring data sets $\mathcal{D} \sim \mathcal{D}'$ in $\mathcal{S}$ and for every measurable event $\mathcal{E} \subseteq \mathcal{X}$,

$$Pr[\mathcal{A}(\mathcal{D}) \in \mathcal{E}] \leqslant e^\epsilon Pr[\mathcal{A}(\mathcal{D}') \in \mathcal{E}] + \vartheta.$$

If $\vartheta = 0$, the algorithm $\mathcal{A}$ satisfies pure $\epsilon$-differential privacy ($\epsilon$-DP).

To simplify privacy analysis, we adopt the CDP, which is defined via the Rényi divergence.

**Definition 2** (Rényi Divergence, Rényi 1961). For two data sets $\mathcal{D}, \mathcal{D}' \subseteq \mathcal{S}$ and a randomized algorithm $\mathcal{A}$, the Rényi divergence of order $\alpha > 1$ between the distributions of $\mathcal{A}(\mathcal{D})$ and $\mathcal{A}(\mathcal{D}')$ is defined as:

$$D_\alpha\big(\mathcal{A}(\mathcal{D})\|\mathcal{A}(\mathcal{D}')\big) := \frac{1}{\alpha-1} \log \int \left(\frac{P_\mathcal{A}(\zeta)}{P_{\mathcal{A}'}(\zeta)}\right)^\alpha P_{\mathcal{A}'}(\zeta)d\zeta,$$

where $P_\mathcal{A}$ and $P_{\mathcal{A}'}$ denote the probability density functions of $\mathcal{A}(\mathcal{D})$ and $\mathcal{A}(\mathcal{D}')$, respectively.

**Definition 3** ($\rho$-CDP, Bun & Steinke 2016). For some $\rho > 0$, a randomized algorithm $\mathcal{A} : \mathcal{S} \to \mathcal{X}$ satisfies $\rho$-CDP if for all pairs of neighboring data sets $\mathcal{D} \sim \mathcal{D}'$ in $\mathcal{S}$ and all orders $\alpha \in (1, +\infty)$, the Rényi divergence between the output distributions satisfies:

$$D_\alpha\big(\mathcal{A}(\mathcal{D})\|\mathcal{A}(\mathcal{D}')\big) \leqslant \alpha\rho.$$

**Definition 4** (Lipschitz-smoothness). A function $F$ is $L$-smooth if for all $x, x' \in \mathbb{R}^d$, $\|\nabla F(x) - \nabla F(x')\|_2 \leqslant L\|x - x'\|_2$.

Note that Lipschitz-smoothness is different from Lipschitz-continuity: the former is defined on the gradient $\nabla F$, while the latter is defined on the function $F$ itself.

**Assumption 1** ($k$-heavy-tailed Assumption). Let $\mathcal{X} \subseteq \mathbb{R}^d$ be a compact convex set, and let $\mathcal{P}$ be a probability distribution over a sample space $\mathcal{S}$. For the convex loss function $f(\cdot; s) : \mathbb{R}^d \to \mathbb{R}$, if there exists a non-decreasing sequence $\{G_j\}_{j=1}^k$ such that for each $j \in [k]$, $\mathbb{E}_{s\sim\mathcal{P}}[\sup_{x\in\mathcal{X}} \|\nabla f(x;s)\|_2^j] \leq G_j^j < \infty$ holds for all $\nabla f(x;s) \in \partial f(x;s)$, then the distribution $\mathcal{P}$ is called $k$-heavy-tailed ($k \geq 2$ in general situations). If $f(\cdot; s)$ is differentiable, the subdifferential $\partial f(x;s)$ (defined in Eq. 12) can be simplified to the gradient $\nabla f(x;s)$.

**Assumption 2** (Lipschitz Assumption). On the basis of $k$-heavy-tailed Assumption, there exists a constant $L_f$ such that $\max_{s\sim\mathcal{P},x\in\mathcal{X}} \|\nabla f(x;s)\|_2 \leqslant L_f < \infty$ holds for all $\nabla f(x;s) \in \partial f(x;s)$.

### 2.2 RELATED WORKS

Under Assumption 1, Lowy & Razaviyayn (2023) develop the LNCSM for privately solving model (1) with non-differentiable $f$. Their method implements a $\log_2(n)$-phase localization scheme, where each phase $t$ solves a regularized empirical subproblem of the following form:

$$\min_{x\in\mathcal{X}_t} \hat{F}_t(x) := \left\{\frac{1}{n_t} \sum_{s_i\in\mathcal{D}_t} f(x; s_i) + \frac{\lambda_t}{2}\|x - x_{t-1}\|_2^2\right\}. \tag{4}$$

For each $t$, $\mathcal{D}_t$ denotes a new batch of $n_t := \frac{n}{2^t}$ i.i.d samples drawn from $\mathcal{P}$, $\lambda_t$ denotes the regularization parameter, and the domain $\mathcal{X}_t := \{x \in \mathcal{X} : \|x - x_{t-1}\|_2 \leqslant \frac{2L_f}{\lambda_t}\}$ is restricted to a neighborhood of the previous solution $x_{t-1}$. Solving the subproblem (4) is challenging, since it is difficult to deal with the constraint $x \in \mathcal{X}_t$. If $f(\cdot; s)$ is $\omega$-strongly convex and $L_f$-Lipschitz continuous (but not necessarily differentiable) for all $s \in \mathcal{S}$, then the output of LNCSM $\hat{x}_{\text{LNCSM}}$ satisfies $\rho$-CDP for $\rho \leqslant \frac{d}{2}$ and achieves the following guarantee:

$$\mathbb{E}[F_{\mathcal{P}}(\hat{x}_{\text{LNCSM}})] - F_{\mathcal{P}}(x^*) \lesssim \frac{G_{2k}^2}{\omega}\left(\frac{1}{n} + \left(\frac{\sqrt{d\log(n)}}{\sqrt{2\rho}n}\right)^{\frac{2k-2}{k}}\right), \tag{5}$$

where the expectation is taken over the random choice of $\mathcal{D}$ and the noise added by LNCSM. Although LNCSM can handle non-differentiable model, it still requires the Lipschitz assumption. Moreover, its computational efficiency is low, which requires $\tilde{O}(n^2 L_f^2/G_{2k}^2)$ subgradient evaluations in total. Besides, the error bound (5) requires $G_{2k} < \infty$, resulting in polynomial EPL in the perspective of $k$-HT setting. Worse still, LNCSM fails to achieve the asymptotic lower bound of EPL in the most popular 2-HT setting (see Table 1). Besides LNCSM, Lowy & Razaviyayn (2023) further propose the NCSGD method for model (1) with smooth and strongly convex $F_{\mathcal{P}}$. This approach incorporates gradient clipping at each iteration to achieve linear-time complexity and attain the optimal EPL bound. Specifically, for model (1) with $L_{\nabla F}$-smooth and $\omega$-strongly convex $F_{\mathcal{P}}$ with $\frac{L_{\nabla F}}{\omega} \leqslant \frac{n}{\log n}$, the output of NCSGD $\hat{x}_{\text{NCSGD}}$ achieves $\rho$-CDP with the following EPL bound:

$$\mathbb{E}[F_{\mathcal{P}}(\hat{x}_{\text{NCSGD}})] - F_{\mathcal{P}}(x^*) \lesssim \frac{G_k^2}{\omega}\left(\frac{1}{n} + \left(\frac{\sqrt{d(L_{\nabla F}/\omega)\log(n)}}{\sqrt{2\rho}n}\right)^{\frac{2k-2}{k}}\right). \tag{6}$$

However, NCSGD is not applicable to NDR.

Asi et al. (2024) further propose the PLLS method to relax the Lipschitz-smoothness condition of NCSGD to a weaker differentiable condition regarding the fidelity $f$. Under Assumption 1, if $f$ is $\omega$-strongly convex and differentiable, then the output of PLLS $\hat{x}_{\text{PLLS}}$ achieves $\rho$-CDP and attains the near-optimal EPL

$$F_{\mathcal{P}}(\hat{x}_{\text{PLLS}}) - F_{\mathcal{P}}(x^*) \lesssim \frac{G_2^2}{\omega} \cdot \frac{\log(1/\sigma)}{n} + \frac{G_k^2}{\omega} \cdot \left(\frac{d\log^3(1/\sigma)}{n^2\rho}\right)^{1-\frac{1}{k}} \tag{7}$$

with a probability of at least $(1 - \sigma)$. To achieve this probabilistic guarantee, they implement an aggregation strategy that results in high computational complexity, requiring $\mathcal{O}\left(\max(n^2, \frac{n^3\rho}{d})\right)$ gradient queries. For model (1) with smooth $f$, Asi et al. (2024) develop a localized-clipped-DP-SGD algorithm based on the localization scheme, gradient clipping, and the sparse vector technique. This method achieves near-linear time complexity but remains inapplicable to NDR.

As shown above, existing state-of-the-art methods cannot efficiently solve model (3) under the HT assumption. This motivates us to develop a computationally tractable algorithm for privately solving this model.

## 3 METHODOLOGY

For convenience, we restate model (3) as the following NDR model for HT DP-SCO, explaining all the involved terms in detail:

$$\min_{x \in \mathcal{X}} \mathcal{F}(x) := \{F_{\mathcal{P}}(x) + g(x)\}. \tag{8}$$

In model (8), $\mathcal{P}$ is an unknown distribution satisfying Assumption 1. For $x \in \mathcal{X}$, the term $F_{\mathcal{P}}(x) := \mathbb{E}_{s\sim\mathcal{P}}[f(x; s)]$ denotes the EPL, where the fidelity $f(\cdot; s) : \mathbb{R}^d \to \mathbb{R}$ is convex. $g(x)$ denotes an NDR term. Given a data set $\mathcal{D} := \{s_1, s_2, \ldots, s_n\} \subseteq \mathcal{S}$ consisting of $n$ i.i.d. samples drawn from $\mathcal{P}$, we aim to privately solve an empirical form of (8)

$$\min_{x \in \mathcal{X}} \{F_{\mathcal{D}}(x) + g(x)\} := \left\{\frac{1}{n}\sum_{s_i \in \mathcal{D}} f(x; s_i) + g(x)\right\}, \tag{9}$$

in order to obtain an approximate solution to the original model (8). We make the following assumptions regarding the empirical fidelity $F_{\mathcal{D}}$ and the NDR $g$.

**Assumption 3.** $g : \mathbb{R}^d \to \mathbb{R}$ is proper, lower semi-continuous, convex, and $L_g$-Lipschitz continuous with some $L_g > 0$.

**Assumption 4.** For the given data set $\mathcal{D}$, $F_{\mathcal{D}} : \mathbb{R}^d \to \mathbb{R}$ is $\omega$-strongly convex with some $\omega > 0$.

**Assumption 5.** For the given data set $\mathcal{D}$, $F_{\mathcal{D}} : \mathbb{R}^d \to \mathbb{R}$ is $L_{\nabla F_{\mathcal{D}}}$-smooth with some $L_{\nabla F_{\mathcal{D}}} > 0$.

These conditions are commonly-used and satisfied by most conventional fidelity and regularization functions. To reformulate model (9) as an unconstrained problem, we introduce the indicator function $\iota_{\mathcal{X}} : \mathbb{R}^d \to \mathbb{R} \cup \{+\infty\}$ for the set $\mathcal{X}$:

$$\iota_{\mathcal{X}}(x) = \begin{cases} 0 & \text{if } x \in \mathcal{X}, \\ +\infty & \text{else.} \end{cases} \tag{10}$$

Then model (9) is equivalent to the following model:

$$\min_{x \in \mathbb{R}^d} \left\{ F_{\mathcal{D}}(x) + g(x) + \iota_{\mathcal{X}}(x) \right\}. \tag{11}$$

Since $g$ and $\iota_{\mathcal{X}}$ are both non-differentiable, model (11) cannot be directly solved by a standard proximal gradient method. It requires careful consideration to construct an efficient solving scheme for model (11). We first solve model (11) non-privately via the forward-backward splitting scheme (Li & Zhang, 2016; Lin et al., 2024a;b;c) in Section 3.1, then extend it to the privacy-preserving setting in Section 3.2.

### 3.1 NON-PRIVATE SETTING

We begin by recalling the definitions of proximity operator, subdifferential, and conjugate function. Let $\Gamma_0(\mathbb{R}^d)$ be the class of proper lower semi-continuous convex functions from $\mathbb{R}^d$ to $\mathbb{R} \cup \{+\infty\}$. The proximity operator of $\psi \in \Gamma_0(\mathbb{R}^d)$ at $x \in \mathbb{R}^d$ is defined as

$$\text{prox}_{\psi}(x) := \arg\min_{v \in \mathbb{R}^d} \left\{ \psi(v) + \frac{1}{2} \|v - x\|_2^2 \right\}.$$

The subdifferential of $\psi \in \Gamma_0(\mathbb{R}^d)$ at $x \in \mathbb{R}^d$ is defined as

$$\partial \psi(x) := \{ y \in \mathbb{R}^d \mid \psi(v) \geqslant \psi(x) + \langle y, v - x \rangle \text{ for all } v \in \mathbb{R}^d \}, \tag{12}$$

where $\langle \cdot, \cdot \rangle$ denotes the inner product. The conjugate function of $\psi$ is defined as

$$\psi^*(x) := \sup_{v \in \mathbb{R}^d} \{ \langle x, v \rangle - \psi(v) \}.$$

We construct an operator $\mathcal{T}_A : \mathbb{R}^{2d} \to \mathbb{R}^{2d}$ such that the solution to model (11) can be characterized by the fixed point set of $\mathcal{T}_A$. Let $z := \begin{bmatrix} x \\ u \end{bmatrix}$ with $x, u \in \mathbb{R}^d$, and define

$$\mathcal{T}_A(z) := \mathcal{K}\left(Az - R\nabla\gamma(z)\right), \tag{13}$$

where

$$A := \begin{bmatrix} I_{[d]} & -\eta I_{[d]} \\ \eta I_{[d]} & I_{[d]} \end{bmatrix}, \quad R := \begin{bmatrix} \eta I_{[d]} & \\ & \eta I_{[d]} \end{bmatrix}. \tag{14}$$

$\eta > 0$ represents learning rate, and $I_{[d]}$ denotes the $d$-dimensional identity matrix. The function $\gamma : \mathbb{R}^{2d} \to \mathbb{R}$ and the operator $\mathcal{K} : \mathbb{R}^{2d} \to \mathbb{R}^{2d}$ are defined as

$$\gamma(z) := F_{\mathcal{D}}(x), \quad \mathcal{K}(z) := \begin{bmatrix} \text{prox}_{\eta g}(x) \\ \text{prox}_{\eta \iota_{\mathcal{X}}^*}(u) \end{bmatrix}. \tag{15}$$

**Theorem 1.** *Let $\mathcal{T}_A$ be defined as (13). Denote the fixed-point set of $\mathcal{T}_A$ by Fix($\mathcal{T}_A$). Under Assumption 3, if $\hat{x}_*$ is a solution to model (11), then for any $\eta > 0$, there exists $\hat{u}_* \in \mathbb{R}^d$ such that $\hat{z}_* = \begin{bmatrix} \hat{x}_* \\ \hat{u}_* \end{bmatrix} \in$ Fix($\mathcal{T}_A$). Conversely, if there exists $\eta > 0$ such that $\hat{z}_* \in$ Fix($\mathcal{T}_A$), then $\hat{x}_*$ is a solution to model (11).*

The proof is provided in Appendix A.1. Theorem 1 implies that solving model (11) is equivalent to finding a fixed point of $\mathcal{T}_A$. However, since the matrix $A$ is expansive, directly solving for Fix($\mathcal{T}_A$) is challenging. To address this problem, we construct a non-expansive operator $\mathcal{T}_C$ by the forward-backward splitting scheme, whose fixed-point set coincides with Fix($\mathcal{T}_A$).

Define operators $\mathcal{T}_Q : \mathbb{R}^{2d} \to \mathbb{R}^{2d}$ and $\mathcal{T}_C : \mathbb{R}^{2d} \to \mathbb{R}^{2d}$ as

$$\mathcal{T}_Q(z) := \{q \in \mathbb{R}^{2d} | q = \mathcal{K}\left((A - Q)q + Qz\right)\} \tag{16}$$

and

$$\mathcal{T}_C := \mathcal{T}_Q \circ (I - C^{-1}\nabla\gamma), \tag{17}$$

where

$$Q := \begin{bmatrix} I_{[d]} & -\eta I_{[d]} \\ -\eta I_{[d]} & I_{[d]} \end{bmatrix}, \quad C := R^{-1}Q = \begin{bmatrix} \frac{1}{\eta}I_{[d]} & -I_{[d]} \\ -I_{[d]} & \frac{1}{\eta}I_{[d]} \end{bmatrix}. \tag{18}$$

The following proposition verifies that the operator $\mathcal{T}_Q$ is well-defined.

**Proposition 1.** *Under Assumption 3, for any given $z \in \mathbb{R}^{2d}$, there exists one unique $q \in \mathbb{R}^{2d}$ such that $\mathcal{T}_Q(z) = q$.*

The proof is provided in Appendix A.2. We now prove the equivalence between the fixed-point sets of $\mathcal{T}_A$ and $\mathcal{T}_C$.

**Proposition 2.** *Let $\mathcal{T}_A$ and $\mathcal{T}_C$ be defined as (13) and (17), respectively. Then Fix($\mathcal{T}_A$) = Fix($\mathcal{T}_C$).*

The proof is provided in Appendix A.3. From Proposition 2, Fix($\mathcal{T}_A$) can be obtained through Fix($\mathcal{T}_C$).

To prove that $\mathcal{T}_C$ is contractive, we introduce the following concepts and properties of non-expansiveness. Let matrix $H \in \mathbb{R}^{d \times d}$ be positive definite. An operator $\mathcal{T} : \mathbb{R}^d \to \mathbb{R}^d$ is non-expansive with respect to (w.r.t.) $H$ if for all $x, y \in \mathbb{R}^d$, $\|\mathcal{T}(x) - \mathcal{T}(y)\|_H \leqslant \|x - y\|_H$, where $\|\cdot\|_H$ denotes the weighted norm defined by $\|x\|_H := \langle x, Hx \rangle^{\frac{1}{2}}$. Further, if $\|\mathcal{T}(x) - \mathcal{T}(y)\|_H^2 \leqslant \langle \mathcal{T}(x) - \mathcal{T}(y), x - y \rangle_H$, then $\mathcal{T}$ is called firmly non-expansive w.r.t. $H$. If there exists a non-expansive operator $\mathcal{N} : \mathbb{R}^d \to \mathbb{R}^d$ w.r.t. $H$ and $\beta \in (0, 1)$ such that $\mathcal{T} = (1 - \beta)\mathcal{I} + \beta\mathcal{N}$, then $T$ is called $\beta$-averaged non-expansive w.r.t. $H$.

We first prove that $\mathcal{K}$ is firmly non-expansive, then establish the contractivity of $\mathcal{T}_C$.

**Proposition 3.** *Let $R$ and $\mathcal{K}$ be defined as (14) and (15), respectively. Under Assumption 3, $\mathcal{K}$ is firmly non-expansive w.r.t. $R^{-1}$.*

The proof is provided in Appendix A.4. It can be easily verified that $C$ is symmetric. Then we can calculate that the maximum and the minimum eigenvalues of $C$ are $\lambda_{\max}(C) = \frac{\eta+1}{\eta} > 0$ and $\lambda_{\min}(C) = \frac{1-\eta}{\eta}$, respectively.

**Proposition 4.** *Let $\mathcal{T}_C$ and $C$ be defined as (17) and (18), respectively. Under Assumptions 3, 4 and 5, if $\eta < \left(1 + \frac{L_{\nabla F_\mathcal{D}}^2}{4\omega} + L_{\nabla F_\mathcal{D}}\sqrt{\frac{L_{\nabla F_\mathcal{D}}^2}{16\omega^2} + \frac{1}{\omega}}\right)^{-1}$, then $\mathcal{T}_C$ is $\kappa$-contractive w.r.t. $C$, where $\kappa := \sqrt{1 - \frac{2\omega\eta}{1+\eta} + \frac{L_{\nabla F_\mathcal{D}}^2\eta^2}{1+\eta^2-2\eta}} \in (0, 1)$.*

The proof is provided in Appendix A.5. Since $\mathcal{T}_C$ is contractive, we can employ a fixed-point iteration of $\mathcal{T}_C$ to find Fix($\mathcal{T}_C$), which can be formulated as follows: for $k \in \mathbb{N}$,

$$z_{(k+1)} = \mathcal{T}_C(z_{(k)}) = \mathcal{T}_Q\left(z_{(k)} - C^{-1}\nabla\gamma(z_{(k)})\right)$$

$$\Leftrightarrow \quad z_{(k+1)} = \mathcal{K}\left((A - Q)z_{(k+1)} + Q\left(z_{(k)} - C^{-1}\nabla\gamma(z_{(k)})\right)\right)$$

$$\Leftrightarrow \quad z_{(k+1)} = \mathcal{K}\left((A - Q)z_{(k+1)} + Qz_{(k)} - R\nabla\gamma(z_{(k)})\right). \tag{19}$$

Let $z_{(k)} = \begin{bmatrix} x_{(k)} \\ u_{(k)} \end{bmatrix}$ with $x_{(k)}, u_{(k)} \in \mathbb{R}^d$. Then (19) can be decomposed into the component-wise updates:

$$\begin{cases} x_{(k+1)} = \text{prox}_{\eta g}\left(x_{(k)} - \eta(\nabla F_\mathcal{D}(x_{(k)}) + u_{(k)})\right) & \text{(20a)} \\ u_{(k+1)} = \text{prox}_{\eta \iota_\mathcal{X}^*}\left(u_{(k)} + \eta(2x_{(k+1)} - x_{(k)})\right). & \text{(20b)} \end{cases}$$

From the relationship between the subdifferential and the proximity operator (see Lemma 2(ii) and (iii) in the appendix), we can reformulate (20b) as follows:

$$u_{(k+1)} \in \partial \iota_{\mathcal{X}} \left( \frac{1}{\eta}(u_{(k)} - u_{(k+1)}) + 2x_{(k+1)} - x_{(k)} \right)$$

$$\Leftrightarrow \frac{1}{\eta}(u_{(k)} - u_{(k+1)}) + 2x_{(k+1)} - x_{(k)} = \text{prox}_{\frac{1}{\eta}\iota_{\mathcal{X}}} \left( \frac{1}{\eta}u_{(k)} + 2x_{(k+1)} - x_{(k)} \right).$$

Hence (20b) can be rewritten as

$$u_{(k+1)} = \eta(\mathcal{I} - \text{prox}_{\frac{1}{\eta}\iota_{\mathcal{X}}}) \left( \frac{1}{\eta}u_{(k)} + 2x_{(k+1)} - x_{(k)} \right). \tag{21}$$

Therefore, the non-private solution to model (11) can be obtained via the joint iterative scheme of (20a) and (21).

## 3.2 PRIVATE SETTING

The privacy-preserving update rule takes the following form: for $k \in \mathbb{N}$,

$$\begin{cases} x_{(k+1)} = \text{prox}_{\eta_{(k+1)}g} \left( \tilde{x}_{(k)} - \eta_{(k+1)} \big( \nabla F_{\mathcal{D}}(\tilde{x}_{(k)}) + \tilde{u}_{(k)} \big) \right), \\ u_{(k+1)} = \eta_{(k+1)} \left( \mathcal{I} - \text{prox}_{\frac{1}{\eta_{(k+1)}}\iota_{\mathcal{X}}} \right) \left( \frac{1}{\eta_{(k+1)}}\tilde{u}_{(k)} + 2x_{(k+1)} - \tilde{x}_{(k)} \right), \\ \begin{bmatrix} \tilde{x}_{(k+1)} \\ \tilde{u}_{(k+1)} \end{bmatrix} = \begin{bmatrix} x_{(k+1)} \\ u_{(k+1)} \end{bmatrix} + \xi_{(k+1)}, \end{cases} \tag{22}$$

where the step size $\eta_{(k)}$ satisfies $\eta_{(k+1)} \leqslant \min \left( \frac{C_{lip}}{n \max(\|\nabla F_{\mathcal{D}}(\tilde{x}_{(k)}) + \tilde{u}_{(k)}\|_2, \|\tilde{x}_{(k)}\|_2)}, \frac{\bar{\eta}}{n} \right)$ with a clipping parameter $C_{lip} > 0$ to ensure the sensitivity bound and an upper bound $\bar{\eta} > 0$ on the step size. The noise $\xi_{(k)} \in \mathbb{R}^{2d}$ is added to $z_{(k)} := \begin{bmatrix} x_{(k)} \\ u_{(k)} \end{bmatrix}$ to preserve the private gradient data $\nabla F_{\mathcal{D}}(x_{(k-1)})$ contained in $z_{(k)}$ for achieving DP.

Let $\tilde{z}_{(k)} := \begin{bmatrix} \tilde{x}_{(k)} \\ \tilde{u}_{(k)} \end{bmatrix}$ with $\tilde{x}_{(k)}, \tilde{u}_{(k)} \in \mathbb{R}^d$. Then (22) can be rewritten as

$$\tilde{z}_{(k+1)} = \mathcal{T}_{C_{(k+1)}}(\tilde{z}_{(k)}; \mathcal{D}) + \xi_{(k+1)}, \tag{23}$$

where the operator $\mathcal{T}_{C_{(k)}}(\cdot; \mathcal{D})$ is defined as (17) applied with the dataset $\mathcal{D}$ and the step size $\eta_{(k)}$. Proposition 5 establishes a utility bound for the iterative scheme (23).

**Definition 5** (Sub-sigma-algebra Flow). Let $\Gamma$ be a sigma-algebra on $\mathbb{R}^{2d}$. Define $\{\Gamma_k\}_{k=0}^K$ as a sub-sigma-algebra flow of $\Gamma$ such that $\Upsilon(\tilde{z}_{(0)}, \ldots, \tilde{z}_{(k-1)}) \subseteq \Gamma_{k-1} \subseteq \Gamma_k$ for all $k$, where $\Upsilon(\tilde{z}_{(0)}, \ldots, \tilde{z}_{(k-1)})$ denotes the sigma-algebra generated by $\tilde{z}_{(0)}, \ldots, \tilde{z}_{(k-1)}$.

**Proposition 5.** *Recall that $\hat{z}_*$ is the fixed-point of $\mathcal{T}_C$. Assume that $\omega > 1$ and let the sequence $\{\tilde{z}_{(k)}\}_{k\in\mathbb{N}}$ be generated by (23) with $\bar{\eta} < \left( 1 + \frac{L_{\nabla F_{\mathcal{D}}}^2}{4\omega - 4} + L_{\nabla F_{\mathcal{D}}} \sqrt{\frac{L_{\nabla F_{\mathcal{D}}}^2}{16(\omega-1)^2} + \frac{1}{w-1}} \right)^{-1}$, where for $k \in \mathbb{N}$, the noise $\xi_{(k)}$ satisfies $\mathbb{E}[\xi_{(k)}] = 0_{[2d]}$ and $\mathbb{E}[\|\xi_{(k)}\|_2^2] = 2d\delta^2$. Under Assumptions 3, 4 and 5, if the sequence $\{\tilde{z}_{(k)}\}_{k=1}^\infty$ is bounded, then there exists a $\bar{\kappa} \in (0,1)$ such that for any $k \in \mathbb{N}$,*

$$\mathbb{E}\left[ \|\tilde{z}_{(k)} - \hat{z}_*\|_2^2 \mid \Gamma_0 \right] \leqslant \bar{\kappa}^{2k} \|\tilde{z}_{(0)} - \hat{z}_*\|_2^2 + \frac{1 - \bar{\kappa}^{2k}}{1 - \bar{\kappa}^2} 2d\delta^2. \tag{24}$$

*where the expectation is taken over the noise added during the iteration.*

The proof is provided in Appendix A.6. The complete algorithm is presented in Algorithm 1, where each iteration incorporates a Gaussian-distributed noise vector.

We prove that for any target privacy level $\rho > 0$, with an appropriately chosen noise variance, the output of Algorithm 1 achieves $\rho$-CDP. The following lemma is essential for this result.

---

**Algorithm 1:** NDR-HT

---

**Require:** Data set $\mathcal{D} := \{s_1, s_2, \ldots, s_n\} \subseteq \mathcal{S}$, fidelity $f : \mathcal{X} \times \mathcal{S} \to \mathbb{R}$, regularization
$\quad g : \mathbb{R}^d \to \mathbb{R}$, and starting point $\tilde{z}_{(0)} \in \mathbb{R}^{2d}$. Set the upper bound of step size $\bar{\eta}$, the clipping
$\quad$ parameter $C_{lip}$, the iteration count $K$, and the noise variance $\delta^2$.
1: **for** $k = 0, 1, 2, \ldots, K$ **do**
2: $\quad$ Sample $\xi_{(k+1)}$ from the Gaussian distribution $\mathcal{N}(0_{[2d]}, \delta^2 I_{[2d]})$.
3: $\quad$ Compute $\tilde{z}_{(k+1)}$ using (23).
4: **end for**
**Ensure:** The solution $\tilde{x}_{(K)}$.

---

**Lemma 1.** *Under Assumption 3, for neighboring data sets $\mathcal{D}, \mathcal{D}' \subseteq \mathcal{S}$ and all $k \in \mathbb{N}$, the operator $\mathcal{T}_{C_{(k)}}(\cdot; \mathcal{D})$ satisfies the sensitivity bound*

$$\|\mathcal{T}_{C_{(k+1)}}(\tilde{z}_{(k)}; \mathcal{D}) - \mathcal{T}_{C_{(k+1)}}(\tilde{z}_{(k)}; \mathcal{D}')\|_2 \leqslant \frac{\theta}{n},$$

*where $\theta := 6C_{lip} + \bar{\eta}L_g + 2\bar{\eta}D + \frac{2C_{lip}\bar{\eta} + 2\bar{\eta}^2 L_g}{n}$ with $D := \max_{x \in \mathcal{X}} \|x\|_2$.*

The proof is provided in Appendix A.7.

**Theorem 2.** *Under Assumption 3, the private solution obtained from Algorithm 1 with $\delta^2 = \frac{K\theta^2}{2n^2\rho}$ achieves $\rho$-CDP, where $\rho > 0$ is the target privacy level.*

The proof is provided in Appendix A.8. The following proposition establishes an error bound between the private solution $\tilde{x}_{(K)}$ obtained from Algorithm 1 and the non-private optimal solution to the empirical model.

**Proposition 6.** *Let $\hat{x}_* \in \mathbb{R}^d$ be the optimal solution to model (11). Under Assumptions 3, 4 and 5, the private solution $\tilde{x}_{(K)}$ obtained from Algorithm 1 with $\delta^2 = \frac{\theta^2}{2n^2\rho}K$ satisfies $\rho$-CDP. Besides, assume that $\omega > 1$ and let $\bar{\eta} < \left(1 + \frac{L^2_{\nabla F_{\mathcal{D}}}}{4\omega - 4} + L_{\nabla F_{\mathcal{D}}}\sqrt{\frac{L^2_{\nabla F_{\mathcal{D}}}}{16(\omega-1)^2} + \frac{1}{w-1}}\right)^{-1}$. If the sequence $\{\tilde{z}_{(k)}\}_{k=1}^{\infty}$ is bounded, then for $K = \bar{\alpha}\log_{\frac{1}{\bar{\kappa}}}(n\sqrt{\rho})$ with $\bar{\alpha} \geqslant 1$,*

$$\mathbb{E}\left[\|\tilde{x}_{(K)} - \hat{x}_*\|_2\right] \leqslant \frac{\tilde{D}}{(n\sqrt{\rho})^{\bar{\alpha}}} + \frac{\theta}{n\sqrt{\rho}}\left[\frac{d\bar{\alpha}}{1 - \bar{\kappa}^2} \cdot \log_{\frac{1}{\bar{\kappa}}}(n\sqrt{\rho})\right]^{\frac{1}{2}},$$

*where $\tilde{D} := 2(D + L_{\nabla F_{\mathcal{D}}}D + L_g)$. Moreover, let function $\mathcal{F}$ be defined in (8). Then under Assumption 1,*

$$\mathbb{E}\left[\left|\mathcal{F}(\tilde{x}_{(K)}) - \mathcal{F}(\hat{x}_*)\right|\right] \leqslant (G_1 + L_g)\left(\frac{\tilde{D}}{(n\sqrt{\rho})^{\bar{\alpha}}} + \frac{\theta}{n\sqrt{\rho}}\left[\frac{d\bar{\alpha}}{1 - \bar{\kappa}^2} \cdot \log_{\frac{1}{\bar{\kappa}}}(n\sqrt{\rho})\right]^{\frac{1}{2}}\right).$$

*The expectation is taken over the noise added during the iteration.*

The proof is provided in Appendix A.9. The following proposition establishes an error bound between the objective function values at the empirical minimizer and the true minimizer of model (8).

**Proposition 7.** *Let $x^*$ and $\hat{x}_*$ be the optimal solutions to model (8) and model (11), respectively. Let $\mathcal{F}$ be defined in (8). Under Assumptions 1, 3 and 4,*

$$\mathbb{E}[\mathcal{F}(\hat{x}_*)] - \mathcal{F}(x^*) \leqslant \frac{4G_2^2 + 2L_g^2}{\omega n}, \tag{25}$$

*where the expectation is taken over the randomness in $\hat{x}_*$ due to the random choice of $\mathcal{D}$.*

The proof is provided in Appendix A.10. Finally, we derive the EPL bound to evaluate the private solution $\tilde{x}_{(K)}$ obtained from Algorithm 1.

**Theorem 3.** *Let $x^*$ be the optimal solution to model (8) and $\mathcal{F}$ be defined in (8). Under Assumptions 1, 3, 4 and 5, the private solution $\tilde{x}_{(K)}$ obtained from Algorithm 1 with $\delta^2 = \frac{\theta^2}{2n^2\rho}K$ satisfies $\rho$-CDP. Besides, assume that $\omega > 1$ and let $\bar{\eta} < \left(1 + \frac{L_{\nabla F_{\mathcal{D}}}^2}{4\omega - 4} + L_{\nabla F_{\mathcal{D}}}\sqrt{\frac{L_{\nabla F_{\mathcal{D}}}^2}{16(\omega-1)^2} + \frac{1}{w-1}}\right)^{-1}$. If the sequence $\{\tilde{z}_{(k)}\}_{k=1}^{\infty}$ is bounded, then for $K = \bar{\alpha}\log_{\frac{1}{\bar{\kappa}}}(n\sqrt{\rho})$ with $\bar{\alpha} \geqslant 1$, Algorithm 1 queries $\bar{\alpha}n\log_{\frac{1}{\bar{\kappa}}}(n\sqrt{\rho})$ sample gradients and satisfies*

$$\mathbb{E}\left[\mathcal{F}(\tilde{x}_{(K)})\right] - \mathcal{F}(x^*)$$
$$\leqslant (G_1 + L_g)\left(\frac{\tilde{D}}{(n\sqrt{\rho})^{\bar{\alpha}}} + \frac{\theta}{n\sqrt{\rho}}\left[\frac{d\bar{\alpha}}{1-\bar{\kappa}^2}\cdot\log_{\frac{1}{\bar{\kappa}}}(n\sqrt{\rho})\right]^{\frac{1}{2}}\right) + \frac{4G_2^2 + 2L_g^2}{\omega n},$$

*where the expectation is taken over the randomness in $\tilde{x}_{(K)}$ due to the random choice of $\mathcal{D}$ and the noise added during the iteration.*

The proof is provided in Appendix A.11. Compared with the theoretical lower EPL bound $\Omega\left(d\left(\frac{1}{n} + \frac{\sqrt{d}}{n\sqrt{\rho}}\right)\right)$ for strongly convex models under the 2-HT setting (Lowy & Razaviyayn, 2023), Theorem 3 indicates that our method attains an asymptotically optimal EPL bound (up to logarithmic factors) by using only $\mathcal{O}(n\log n)$ gradient queries.

## 4 EXPERIMENTS

We evaluate the performance of NDR-HT through three experiments: (1) linear regression tasks on synthetic data and real-world data from Kaggle[1], and (2) a logistic regression task on real-world data from the UCI Machine Learning Repository[2]. **Further details on evaluation model, data processing and parameter settings are provided in Appendix A.12 and A.13.** The performance of NDR-HT is examined under different privacy budgets. According to Theorem 3, the privacy budget is measured in terms of $\rho$-CDP to facilitate comparison with other methods. Eight distinct values of $\rho_{CDP}$ are evenly spread from 0.01 to 10 in the logarithmic scale. It covers a wide range of privacy levels from low $\rho_{CDP}$ to high $\rho_{CDP}$.

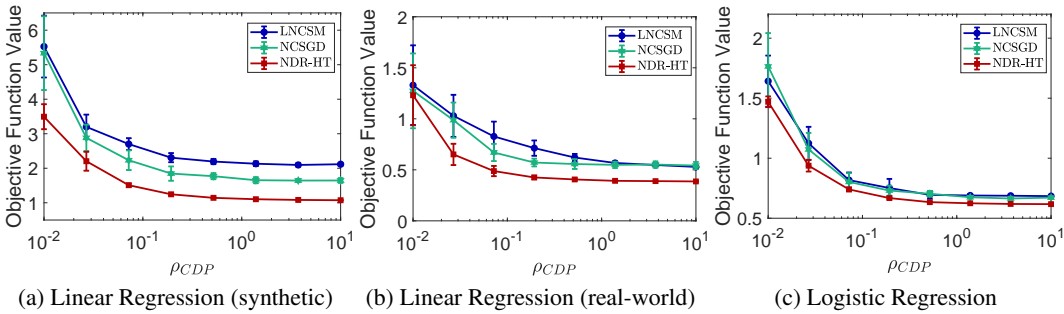

(a) Linear Regression (synthetic)    (b) Linear Regression (real-world)    (c) Logistic Regression

Figure 1: Final objective function values (mean $\pm$ STD) of LNCSM, NCSGD and NDR-HT.

PLLS (Asi et al., 2024) is an oracle calculation framework and does not provide an implementable scheme for numerical experiments (neither does it present numerical experiments in its original paper), while LNCSM and NCSGD (Lowy & Razaviyayn, 2023) are implementable for numerical experiments. Hence we evaluate the performance of NDR-HT against LNCSM and NCSGD by comparing the mean and standard deviation (STD) of the final objective values ($\{F_{\mathcal{D}}(x) + g(x)\}$ in Eq. 9). For LNCSM, we use the same regularization parameters as in our method to ensure a fair comparison. For NCSGD, we set the regularization parameter for NDR to 0, as it cannot be applied to NDR. As shown in Figure 1, NDR-HT consistently outperforms both baselines across

---

[1]https://www.kaggle.com/c/house-prices-advanced-regression-techniques
[2]https://archive.ics.uci.edu/dataset/2/adult

all privacy budgets in both linear and logistic regression experiments. In contrast, NCSGD fails to optimize non-differentiable objectives. Although LNCSM supports such models, it incurs higher computational complexity (see Table A5). **Additional experimental results are provided in Appendix A.14,** which show the effectiveness and efficiency of our method.

## 5 CONCLUSIONS AND FUTURE WORKS

We propose a novel forward-backward splitting approach called NDR-HT to tackle non-differentiable regularization (NDR) for heavy-tailed (HT) differentially private stochastic convex optimization (DP-SCO). It addresses the main limitations of existing methods, such as high excess population loss (EPL), high computational complexity, and inapplicable to NDR. NDR-HT achieves $\rho$-CDP and an asymptotically optimal EPL bound (up to logarithmic factors), and requires only $\mathcal{O}(n \log n)$ gradient evaluations. Moreover, it achieves linear convergence up to an additive approximation error. It is straightforward to implement and free of solving subproblems within iterations.

Experimental results on both synthetic and real-world data show that NDR-HT consistently outperforms existing state-of-the-art approaches under varying privacy budgets, which validate the effectiveness and efficiency of NDR-HT. Future works may fall into extensions to federated learning settings or other classes of non-smooth DP-SCO problems.

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

# A APPENDIX

## A.1 PROOF OF THEOREM 1

**Lemma 2** ([Rockafellar & Wets 2009](#)). *Let $\psi \in \Gamma_0(\mathbb{R}^d)$. Then the following facts hold:*

*(i) (Fermat's rule). $x$ is a minimizer of $\psi$ if and only if $0 \in \partial\psi(x)$.*

*(ii) $y \in \partial\psi(x)$ if and only if $x = \mathrm{prox}_\psi(x + y)$.*

*(iii) $y \in \partial\psi(x)$ if and only if $x \in \partial\psi^*(y)$.*

*(iv) For any $x \in \mathbb{R}^d$, $\mathrm{prox}_\psi(x)$ exists and is unique.*

*(v) $\psi^* \in \Gamma_0(\mathbb{R}^n)$ and $\psi^{**} = \psi$.*

*Proof of Theorem 1.* We know from Fermat's rule and the additivity property of subdifferential that
$$0_d \in \nabla F_{\mathcal{D}}(\hat{x}_*) + \partial g(\hat{x}_*) + \partial\iota_{\mathcal{X}}(\hat{x}_*). \tag{26}$$
Then for $\eta > 0$,
$$-\eta\nabla F_{\mathcal{D}}(\hat{x}_*) \in \eta\partial g(\hat{x}_*) + \eta\partial\iota_{\mathcal{X}}(\hat{x}_*).$$
Thus, there exists
$$\hat{u}_* \in \partial\iota_{\mathcal{X}}(\hat{x}_*) \tag{27}$$
satisfying
$$-\eta(\nabla F_{\mathcal{D}}(\hat{x}_*) + \hat{u}_*) \in \eta\partial g(\hat{x}_*). \tag{28}$$
From Lemma 2 (iii) and (27), we know that
$$\eta\hat{x}_* \in \eta\partial\iota_{\mathcal{X}}^*(\hat{u}_*). \tag{29}$$
Then we can deduce from Lemma 2 (ii), (28) and (29) that
$$\hat{x}_* = \mathrm{prox}_{\eta g}\left(\hat{x}_* - \eta(\nabla F_{\mathcal{D}}(\hat{x}_*) + \hat{u}_*)\right), \tag{30}$$
$$\hat{u}_* = \mathrm{prox}_{\eta\iota_{\mathcal{X}}^*}(\hat{u}_* + \eta\hat{x}_*), \tag{31}$$
which together with the definition of $\mathcal{T}_A$ in (13) implies that $\hat{z}_* = \mathcal{T}_A(\hat{z}_*)$.

Conversely, suppose $\hat{z}_*$ is a fixed-point of $\mathcal{T}_A$ for some $\eta > 0$. Then by the definition of $\mathcal{T}_A$, we know that (30) and (31) hold, which indicates that (27) and (28) hold, and thus (26) holds. Then it follows from Fermat's rule that $\hat{x}_*$ is a solution to model (11). □

## A.2 PROOF OF PROPOSITION 1

*Proof.* Let $q = \begin{bmatrix} q_1 \\ q_2 \end{bmatrix}$ with $q_1, q_2 \in \mathbb{R}^d$. By the definition of $\mathcal{T}_Q$, for a given $z = \begin{bmatrix} x \\ u \end{bmatrix}$, we need to verify that there exists a unique $q \in \mathbb{R}^{2d}$ that satisfies
$$\begin{cases} q_1 = \mathrm{prox}_{\eta g}(x - \eta u), & \text{(32a)} \\ q_2 = \mathrm{prox}_{\eta\iota_{\mathcal{X}}^*}(2\eta q_1 - \eta x + u). & \text{(32b)} \end{cases}$$
Since $g \in \Gamma_0(\mathbb{R}^d)$, it follows from Lemma 2 (iv) that $q_1$ exists and is unique. It can be easily derived that $\iota_{\mathcal{X}} \in \Gamma_0(\mathbb{R}^d)$. Then from Lemma 2 (v), $\iota_{\mathcal{X}}^* \in \Gamma_0(\mathbb{R}^d)$. Thus from Lemma 2 (iv) again, $q_2$ exists and is unique. In summary, $q$ exists and is unique. □

## A.3 PROOF OF PROPOSITION 2

*Proof.* From the definitions of $\mathcal{T}_A$ and $\mathcal{T}_C$, we can deduce that
$$z \in \mathrm{Fix}(\mathcal{T}_C) \Leftrightarrow z = \mathcal{T}_Q(z - C^{-1}\nabla\gamma(z))$$
$$\Leftrightarrow z = \mathcal{K}((A - Q)z + Q(z - C^{-1}\nabla\gamma(z)))$$
$$\Leftrightarrow z \in \mathrm{Fix}(\mathcal{T}_A).$$
The third equivalence holds from $QC^{-1} = R$, which can be directly obtained from the definition of $C$ in (18). □

### A.4 PROOF OF PROPOSITION 3

**Lemma 3** (Combettes & Yamada 2015). *Let $\psi \in \Gamma_0(\mathbb{R}^d)$. Then $\mathrm{prox}_\psi$ is firmly non-expansive w.r.t. $I$.*

*Proof of Proposition 3.* Since $\eta g \in \Gamma_0(\mathbb{R}^d)$ and $\eta \iota_{\mathcal{X}}^* \in \Gamma_0(\mathbb{R}^d)$, it follows from Lemma 3 that $\mathrm{prox}_{\eta g}$ and $\mathrm{prox}_{\eta \iota_{\mathcal{X}}^*}$ are both firmly non-expansive w.r.t. $I$. For $t = \begin{bmatrix} t_1 \\ t_2 \end{bmatrix}$ with $t_1, t_2 \in \mathbb{R}^d$ and $r = \begin{bmatrix} r_1 \\ r_2 \end{bmatrix}$ with $r_1, r_2 \in \mathbb{R}^d$, let

$$p_1 := \mathrm{prox}_{\eta g}(t_1) - \mathrm{prox}_{\eta g}(r_1),$$
$$p_2 := \mathrm{prox}_{\eta \iota_{\mathcal{X}}^*}(t_2) - \mathrm{prox}_{\eta \iota_{\mathcal{X}}^*}(r_2),$$

and $p = \begin{bmatrix} p_1 \\ p_2 \end{bmatrix}$. Then $p = \mathcal{K}(t) - \mathcal{K}(r)$. From the firmly non-expansiveness of $\mathrm{prox}_{\eta g}$ and $\mathrm{prox}_{\eta \iota_{\mathcal{X}}^*}$, we know that

$$\|p_1\|_2^2 \le \langle p_1, t_1 - r_1 \rangle \quad \text{and} \quad \|p_2\|_2^2 \le \langle p_2, t_2 - r_2 \rangle.$$

Then

$$\|\mathcal{K}(t) - \mathcal{K}(r)\|_{R^{-1}}^2 = \|p\|_{R^{-1}}^2 = \frac{1}{\eta}\|p_1\|_2^2 + \frac{1}{\eta}\|p_2\|_2^2$$

$$\le \frac{1}{\eta}\langle p_1, t_1 - r_1 \rangle + \frac{1}{\eta}\langle p_2, t_2 - r_2 \rangle$$

$$= \langle p, t - r \rangle_{R^{-1}} = \langle \mathcal{K}(t) - \mathcal{K}(r), t - r \rangle_{R^{-1}},$$

which implies that $\mathcal{K}$ is firmly non-expansive w.r.t. $R^{-1}$. $\qquad\square$

### A.5 PROOF OF PROPOSITION 4

*Proof.* We know from $\lambda_{\min}(C) = \frac{1-\eta}{\eta}$ and $0 < \eta < 1$ that $C$ is positive definite.

We begin by verifying the non-expansiveness of $\mathcal{T}_Q$. Let $t = \mathcal{T}_Q(c)$, $h = \mathcal{T}_Q(\phi)$ for $c, \phi \in \mathbb{R}^{2d}$, and $a = Q(c - t)$, $b = Q(\phi - h)$. Then

$$\begin{cases} t = \mathcal{K}\left((A - Q)t + Qc\right) = \mathcal{K}\left(At + a\right), \\ h = \mathcal{K}\left((A - Q)h + Q\phi\right) = \mathcal{K}\left(Ah + b\right). \end{cases} \tag{33}$$

From Proposition 3, we know that $\mathcal{K}$ is firmly nonexpansive w.r.t. $R^{-1}$, which together with (33) yields

$$\|t - h\|_{R^{-1}}^2 \le \langle t - h, A(t - h) + (a - b) \rangle_{R^{-1}}. \tag{34}$$

It is equivalent to

$$\langle t - h, a - b \rangle_{R^{-1}} \ge \langle t - h, \tilde{A}(t - h) \rangle, \tag{35}$$

where $\tilde{A} := R^{-1}(I - A) = \begin{pmatrix} 0 & I \\ -I & 0 \end{pmatrix}$. Note that $\tilde{A} = -\tilde{A}^\top$. Then

$$\langle t - h, \tilde{A}(t - h) \rangle = (t - h)^\top \tilde{A}(t - h) = -(t - h)^\top \tilde{A}^\top(t - h) = -\langle \tilde{A}(t - h), (t - h) \rangle, \tag{36}$$

which implies that $\langle \tilde{A}(t - h), (t - h) \rangle = 0$. Then (35) becomes

$$\langle t - h, a - b \rangle_{R^{-1}} \ge 0. \tag{37}$$

Recall that $C := R^{-1}Q$. Then from the definitions of $a$ and $b$,

$$\|t - h\|_C^2 \le \langle t - h, c - \phi \rangle_C. \tag{38}$$

Since $C$ is positive definite, then there exists a matrix $B \in \mathbb{R}^{2d}$ such that $C = B^\top B$. Then we know from (38) that

$$
\begin{aligned}
\|B(t-h)\|_2^2 = \langle B(t-h), B(t-h) \rangle = \langle t-h, C(t-h) \rangle = \|t-h\|_C^2 \\
\leqslant \langle t-h, c-\phi \rangle_C = \langle B(t-h), B(c-\phi) \rangle \leqslant \|B(t-h)\|_2 \|B(c-\phi)\|_2,
\end{aligned}
$$

which implies $\|B(t-h)\|_2^2 \leqslant \|B(c-\phi)\|_2^2$. Therefore,

$$
\|t-h\|_C^2 \leqslant \|c-\phi\|_C^2, \tag{39}
$$

which implies that $\mathcal{T}_Q$ is non-expansive w.r.t. $C$.

We then prove that the operator $\mathcal{I} - C^{-1}\nabla\gamma$ is strictly contractive w.r.t. $C$. For $z, w \in \mathbb{R}^{2d}$, define $l = \nabla\gamma(z) - \nabla\gamma(w)$. Then

$$
\begin{aligned}
&\|(\mathcal{I} - C^{-1}\nabla\gamma)(z) - (\mathcal{I} - C^{-1}\nabla\gamma)(w)\|_C^2 \\
=&\|(z-w) - C^{-1}l\|_C^2 \\
=&\|z-w\|_C^2 - 2\langle z-w, l \rangle + \|C^{-1}l\|_C^2.
\end{aligned} \tag{40}
$$

Recall that $\gamma$ is $\omega$-strongly convex. Then

$$
\langle z-w, l \rangle \geqslant \omega \|z-w\|_2^2. \tag{41}
$$

It also holds from the $L_{\nabla F_\mathcal{D}}$-Lipschitz continuity of $\nabla\gamma(z)$ that

$$
\|C^{-1}l\|_C^2 = l^\top (C^{-1})^\top l = l^\top C^{-1} l \leqslant \frac{1}{\lambda_{\min}(C)} \|l\|_2^2 \leqslant \frac{L_{\nabla F_\mathcal{D}}^2}{\lambda_{\min}(C)} \|z-w\|_2^2. \tag{42}
$$

Combining (40), (41) and (42), we have

$$
\begin{aligned}
&\|(\mathcal{I} - C^{-1}\nabla\gamma)(z) - (\mathcal{I} - C^{-1}\nabla\gamma)(w)\|_C^2 \\
\leqslant&\|z-w\|_C^2 - 2\omega\|z-w\|_2^2 + \frac{L_{\nabla F_\mathcal{D}}^2}{\lambda_{\min}(C)} \|z-w\|_2^2 \\
\leqslant&\left( 1 - \frac{2\omega}{\lambda_{\max}(C)} + \frac{L_{\nabla F_\mathcal{D}}^2}{\lambda_{\min}^2(C)} \right) \|z-w\|_C^2.
\end{aligned} \tag{43}
$$

Let $\kappa := \sqrt{1 - \frac{2\omega}{\lambda_{\max}(C)} + \frac{L_{\nabla F_\mathcal{D}}^2}{\lambda_{\min}^2(C)}}$. Then we need to verify that

$$
0 \leqslant 1 - \frac{2\omega}{\lambda_{\max}(C)} + \frac{L_{\nabla F_\mathcal{D}}^2}{\lambda_{\min}^2(C)} < 1. \tag{44}
$$

Since $L_{\nabla F_\mathcal{D}} \geqslant \omega$, $\lambda_{\min} + \frac{L_{\nabla F_\mathcal{D}}^2}{\lambda_{\min}(C)} \geqslant 2L_{\nabla F_\mathcal{D}} \geqslant 2\omega$, which implies that

$$
\begin{aligned}
&1 - \frac{2\omega}{\lambda_{\max}(C)} + \frac{L_{\nabla F_\mathcal{D}}^2}{\lambda_{\min}^2(C)} \\
\geqslant&1 - \frac{2\omega}{\lambda_{\min}(C)} + \frac{L_{\nabla F_\mathcal{D}}^2}{\lambda_{\min}^2(C)} \\
=&\frac{1}{\lambda_{\min}(C)} \left( \lambda_{\min}(C) + \frac{L_{\nabla F_\mathcal{D}}^2}{\lambda_{\min}(C)} - 2\omega \right) \geqslant 0.
\end{aligned}
$$

Then the left side of (44) holds. It can be easily deduced that the right side of (44) holds if $\lambda_{\min}(C) > \sqrt{\frac{\lambda_{\max}(C)}{2\omega}} L_{\nabla F_\mathcal{D}}$. Thus, the operator $\mathcal{I} - C^{-1}\nabla\gamma$ with $\lambda_{\min}(C) > \sqrt{\frac{\lambda_{\max}(C)}{2\omega}} L_{\nabla F_\mathcal{D}}$ is strictly contractive w.r.t. $C$.

Let $c = (\mathcal{I} - C^{-1}\nabla\gamma)(z)$ and $\phi = (\mathcal{I} - C^{-1}\nabla\gamma)(w)$. Then it follows from (39) and (43) that

$$
\|\mathcal{T}_Q \left( (\mathcal{I} - C^{-1}\nabla\gamma)(z) \right) - \mathcal{T}_Q \left( (\mathcal{I} - C^{-1}\nabla\gamma)(w) \right)\|_C^2 \leqslant \|c-\phi\|_C^2 \leqslant \kappa^2 \|z-w\|_C^2. \tag{45}
$$

Recall that $\lambda_{\max}(C) = \frac{1+\eta}{\eta}$ and $\lambda_{\min}(C) = \frac{1-\eta}{\eta}$. Then $\kappa = \sqrt{1 - \frac{2\omega\eta}{1+\eta} + \frac{L_{\nabla F_{\mathcal{D}}}^2 \eta^2}{1-2\eta+\eta^2}}$. Now we verify that

$$\frac{1}{\eta} - 1 > \sqrt{\frac{\frac{1}{\eta}+1}{2\omega}} L_{\nabla F_{\mathcal{D}}}. \tag{46}$$

Let $\hat{\eta} := \frac{1}{\eta} > 1$. Then (46) is equivalent to

$$(\hat{\eta} - 1)^2 > \frac{L_{\nabla F_{\mathcal{D}}}^2}{2\omega}(\hat{\eta} + 1)$$

$$\Leftrightarrow \eta^2 - (2 + \frac{L_{\nabla F_{\mathcal{D}}}^2}{2\omega})\hat{\eta} + 1 - \frac{L_{\nabla F_{\mathcal{D}}}^2}{2\omega} > 0,$$

which holds for $\hat{\eta} > 1 + \frac{L_{\nabla F_{\mathcal{D}}}^2}{4\omega} + \sqrt{\frac{L_{\nabla F_{\mathcal{D}}}^4}{16\omega^2} + \frac{L_{\nabla F_{\mathcal{D}}}^2}{\omega}}$. This completes the proof.

$\square$

### A.6 PROOF OF PROPOSITION 5

*Proof.* For notational simplicity, we denote the operator $\mathcal{T}_{C(k)}(\cdot; \mathcal{D})$ by $\mathcal{T}_{C(k)}(\cdot)$ throughout this proof. From Proposition 4, for any $k \in \mathbb{N}$, let

$$\kappa_{(k)} := \sqrt{1 - \frac{2\omega\eta_{(k)}}{1 + \eta_{(k)}} + \frac{L_{\nabla F_{\mathcal{D}}}^2 \eta_{(k)}^2}{1 + \eta_{(k)}^2 - 2\eta_{(k)}}}.$$

Then

$$\|\mathcal{T}_{C_{(k+1)}}(\tilde{z}_{(k)}) - \hat{z}_*\|_{C_{(k+1)}}^2 \leqslant \kappa_{(k+1)}^2 \|\tilde{z}_{(k)} - \hat{z}_*\|_{C_{(k+1)}}^2,$$

which indicates that

$$\|\mathcal{T}_{C_{(k+1)}}(\tilde{z}_{(k)}) - \hat{z}_*\|_2^2 \leqslant \frac{1 + \eta_{(k+1)}}{1 - \eta_{(k+1)}} \left(1 - \frac{2\omega\eta_{(k+1)}}{1 + \eta_{(k+1)}} + \frac{L_{\nabla F_{\mathcal{D}}}^2 \eta_{(k+1)}^2}{1 + \eta_{(k+1)}^2 - 2\eta_{(k+1)}}\right) \|\tilde{z}_{(k)} - \hat{z}_*\|_2^2,$$

We verify that for $0 < \eta_{(k)} < \bar{\eta} := \frac{1}{1 + \frac{L_{\nabla F_{\mathcal{D}}}^2}{4\omega - 4} + L_{\nabla F_{\mathcal{D}}}\sqrt{\frac{L_{\nabla F_{\mathcal{D}}}^2}{16(\omega-1)^2} + \frac{1}{w-1}}}$,

$$\frac{1 + \eta_{(k)}}{1 - \eta_{(k)}} \left(1 - \frac{2\omega\eta_{(k)}}{1 + \eta_{(k)}} + \frac{L_{\nabla F_{\mathcal{D}}}^2 \eta_{(k)}^2}{1 + \eta_{(k)}^2 - 2\eta_{(k)}}\right) = 1 - \frac{(2\omega - 2)\eta_{(k)}}{1 - \eta_{(k)}} + \frac{L_{\nabla F_{\mathcal{D}}}^2 \eta_{(k)}^2 (1 + \eta_{(k)})}{(1 - \eta_{(k)})^3} < 1.$$

For $0 < \eta_{(k)} < 1$, the above inequality is equivalent to

$$(2 - 2\omega) + \frac{L_{\nabla F_{\mathcal{D}}}^2 \eta_{(k)}(1 + \eta_{(k)})}{(1 - \eta_{(k)})^2} < 0$$

$$\Leftrightarrow (2 - 2\omega)(1 - \eta_{(k)})^2 + L_{\nabla F_{\mathcal{D}}}^2 \eta_{(k)}(1 + \eta_{(k)}) < 0$$

$$\Leftrightarrow \left(2 - 2\omega + L_{\nabla F_{\mathcal{D}}}^2\right)\eta_{(k)}^2 + \left(L_{\nabla F_{\mathcal{D}}}^2 - 2(2 - 2\omega)\right)\eta_{(k)} + (2 - 2\omega) < 0. \tag{47}$$

If $2 - 2\omega + L_{\nabla F_{\mathcal{D}}}^2 \leqslant 0$, then $\omega \geqslant 1 + \frac{L_{\nabla F_{\mathcal{D}}}^2}{2}$, which contradicts the fact that $L_{\nabla F_{\mathcal{D}}} \geqslant \omega$. Therefore, $2 - 2\omega + L_{\nabla F_{\mathcal{D}}}^2 > 0$. Let $\omega > 1$. Then (47) holds for $0 < \eta_{(k)} < \bar{\eta}$.

If the sequence $\{\tilde{z}_{(k)}\}_{k=1}^{\infty}$ is bounded, then $\{\nabla F_{\mathcal{D}}(\tilde{z}_{(k)})\}_{k=1}^{\infty}$ is bounded due to the continuity of $\nabla F_{\mathcal{D}}$. Then $\{\eta_{(k)}\}_{k=1}^{\infty}$ is bounded, which implies that there exist $\eta_1, \eta_2 > 0$ such that for any $k \in \mathbb{N}$, $\eta_1 \leqslant \eta_{(k)} \leqslant \eta_2 < \bar{\eta}$. Let $\hat{\kappa}_{(k)} := \sqrt{1 - \frac{(2\omega-2)\eta_{(k)}}{1-\eta_{(k)}} + \frac{L_{\nabla F_{\mathcal{D}}}^2 \eta_{(k)}^2 (1+\eta_{(k)})}{(1-\eta_{(k)})^3}} < 1$. Then there exists a $\bar{\kappa} \in (0, 1)$ such that for any $k \in \mathbb{N}$, $\hat{\kappa}_{(k)} \leqslant \bar{\kappa}$. Then for any $k \in \mathbb{N}$,

$$\|\mathcal{T}_{C_{(k+1)}}(\tilde{z}_{(k)}) - \hat{z}_*\|_2 \leqslant \bar{\kappa}\|\tilde{z}_{(k)} - \hat{z}_*\|_2. \tag{48}$$

From (23), we know that

$$\|\tilde{z}_{(k+1)} - \hat{z}_*\|_2^2 = \|\mathcal{T}_{C_{(k+1)}}(\tilde{z}_{(k)}) - \hat{z}_*\|_2^2 + 2\langle\mathcal{T}_{C_{(k+1)}}(\tilde{z}_{(k)}) - \hat{z}_*, \xi_{(k+1)}\rangle + \|\xi_{(k+1)}\|_2^2. \quad (49)$$

Combining (48) and (49), we have

$$\mathbb{E}[\|\tilde{z}_{(k+1)} - \hat{z}_*\|_2^2|\Gamma_k]$$
$$=\mathbb{E}[\|\mathcal{T}_{C_{(k+1)}}(\tilde{z}_{(k)}) - \hat{z}_*\|_2^2 \mid \Gamma_k] + 2\mathbb{E}\left[\langle\mathcal{T}_{C_{(k+1)}}(\tilde{z}_{(k)}) - \hat{z}_*, \xi_{(k+1)}\rangle \mid \Gamma_k\right] + \mathbb{E}[\|\xi_{(k+1)}\|_2^2 \mid \Gamma_k]$$
$$=\|\mathcal{T}_{C_{(k+1)}}(\tilde{z}_{(k)}) - \hat{z}_*\|_2^2 + 2(\mathcal{T}_{C_{(k+1)}}(\tilde{z}_{(k)}) - \hat{z}_*)^\top\mathbb{E}[\xi_{(k+1)} \mid \Gamma_k] + \mathbb{E}[\|\xi_{(k+1)}\|_2^2 \mid \Gamma_k]$$
$$=\|\mathcal{T}_{C_{(k+1)}}(\tilde{z}_{(k)}) - \hat{z}_*\|_2^2 + \mathbb{E}[\|\xi_{(k+1)}\|_2^2 \mid \Gamma_k]$$
$$\leqslant\bar{\kappa}^2\|\tilde{z}_{(k)} - \hat{z}_*\|_2^2 + \varsigma^2, \quad (50)$$

where $\varsigma^2 = 2d\delta^2$. The third equality holds from $\mathbb{E}[\xi_{(k+1)}] = 0_{[2d]}$, where $0_{[2d]}$ denotes the $2d$-dimensional zero vector, and the last inequality follows from $\mathbb{E}[\|\xi_{(k+1)}\|_2^2] \leqslant 2d\delta^2$. Taking expectations conditioned on $\Gamma_{k-1}$ in both sides of (50) yields

$$\mathbb{E}\left[\mathbb{E}[\|\tilde{z}_{(k+1)} - \hat{z}_*\|_2^2|\Gamma_k]|\Gamma_{k-1}\right] \leqslant \bar{\kappa}^2\mathbb{E}[\|\tilde{z}_{(k)} - \hat{z}_*\|_2^2|\Gamma_{k-1}] + \varsigma^2. \quad (51)$$

Since $\Gamma_{k-1} \subseteq \Gamma_k$, then $\mathbb{E}\left[\mathbb{E}[\|\tilde{z}_{(k+1)} - \hat{z}_*\|_2^2|\Gamma_k]|\Gamma_{k-1}\right] = \mathbb{E}[\|\tilde{z}_{(k+1)} - \hat{z}_*\|_2^2|\Gamma_{k-1}]$. Then (51) can be simplified as

$$\mathbb{E}[\|\tilde{z}_{(k+1)} - \hat{z}_*\|_2^2|\Gamma_{k-1}] \leqslant \bar{\kappa}^2\mathbb{E}[\|\tilde{z}_{(k)} - \hat{z}_*\|_2^2|\Gamma_{k-1}] + \varsigma^2. \quad (52)$$

Similarly, successively taking expectations conditioned on $\Gamma_{k-2}, \Gamma_{k-3}, \ldots, \Gamma_0$ yields

$$\mathbb{E}[\|\tilde{z}_{(k+1)} - \hat{z}_*\|_2^2|\Gamma_0] \leqslant \bar{\kappa}^2\mathbb{E}[\|\tilde{z}_{(k)} - \hat{z}_*\|_2^2|\Gamma_0] + \varsigma^2.$$

Since $0 < \bar{\kappa} < 1$, then

$$\mathbb{E}[\|\tilde{z}_{(k+1)} - \hat{z}_*\|_2^2 \mid \Gamma_0]$$
$$\leqslant\bar{\kappa}^2\mathbb{E}[\|\tilde{z}_{(k)} - \hat{z}_*\|_2^2|\Gamma_0] + \varsigma^2$$
$$\leqslant\bar{\kappa}^4\mathbb{E}[\|\tilde{z}_{(k-1)} - \hat{z}_*\|_2^2|\Gamma_0] + (1 + \bar{\kappa}^2)\varsigma^2$$
$$\leqslant\ldots$$
$$\leqslant\bar{\kappa}^{2(k+1)}\|\tilde{z}_{(0)} - \hat{z}_*\|_2^2 + \varsigma^2\sum_{j=0}^{k}\bar{\kappa}^{2j}$$
$$=\bar{\kappa}^{2(k+1)}\|\tilde{z}_{(0)} - \hat{z}_*\|_2^2 + \frac{1 - \bar{\kappa}^{2(k+1)}}{1 - \bar{\kappa}^2}\varsigma^2, \quad (53)$$

which completes the proof. $\qquad\square$

### A.7 PROOF OF LEMMA 1

*Proof.* From (22), we can deduce that

$$\|x_{(k+1)} - x'_{(k+1)}\|_2 \quad (54)$$
$$=\|\text{prox}_{\eta_{(k+1)}g}\left(\tilde{x}_{(k)}-\eta_{(k+1)}\left(\nabla F_\mathcal{D}(\tilde{x}_{(k)})+\tilde{u}_{(k)}\right)\right) - \text{prox}_{\eta'_{(k+1)}g}(\tilde{x}_{(k)}-\eta'_{(k+1)}\left(\nabla F_{\mathcal{D}'}(\tilde{x}_{(k)})+\tilde{u}_{(k)}\right))\|_2$$
$$\leqslant\|\text{prox}_{\eta_{(k+1)}g}\left(\tilde{x}_{(k)}-\eta_{(k+1)}(\nabla F_\mathcal{D}(\tilde{x}_{(k)})+\tilde{u}_{(k)})\right) - \text{prox}_{\eta_{(k+1)}g}\left(\tilde{x}_{(k)}-\eta'_{(k+1)}(\nabla F_{\mathcal{D}'}(\tilde{x}_{(k)})+\tilde{u}_{(k)})\right)\|_2$$
$$+\|\text{prox}_{\eta_{(k+1)}g}\left(\tilde{x}_{(k)}-\eta'_{(k+1)}(\nabla F_{\mathcal{D}'}(\tilde{x}_{(k)})+\tilde{u}_{(k)})\right) - \text{prox}_{\eta'_{(k+1)}g}\left(\tilde{x}_{(k)}-\eta'_{(k+1)}(\nabla F_{\mathcal{D}'}(\tilde{x}_{(k)})+\tilde{u}_{(k)})\right)\|_2$$
$$\leqslant\|\eta_{(k+1)}(\nabla F_\mathcal{D}(\tilde{x}_{(k)}) + \tilde{u}_{(k)}) - \eta'_{(k+1)}(\nabla F_{\mathcal{D}'}(\tilde{x}_{(k)}) + \tilde{u}_{(k)})\|_2 + \bar{\eta}L_g$$
$$\leqslant\frac{2C_{lip}}{n} + \frac{\bar{\eta}L_g}{n}. \quad (55)$$

From the definition of $\text{prox}_{\eta_{(k)}g}$, we know that

$$x_{(k+1)} = \tilde{x}_{(k)}-\eta_{(k+1)}(\nabla F_\mathcal{D}(\tilde{x}_{(k)}) + \tilde{u}_{(k)}) - \eta_{(k+1)}\partial g(x_{(k+1)}),$$
$$x'_{(k+1)} = \tilde{x}_{(k)}-\eta'_{(k+1)}(\nabla F_{\mathcal{D}'}(\tilde{x}_{(k)}) + \tilde{u}_{(k)}) - \eta'_{(k+1)}\partial g(x'_{(k+1)}).$$

Then

$$\|\eta_{(k+1)}\tilde{x}_{(k+1)} - \eta'_{(k+1)}\tilde{x}'_{(k+1)}\|_2$$

$$=\|(\eta_{(k+1)} - \eta'_{(k+1)})\tilde{x}_{(k)} + \eta'^2_{(k+1)}(\nabla F_{\mathcal{D}'}(\tilde{x}_{(k)}) + \tilde{u}_{(k)}) - \eta^2_{(k+1)}(\nabla F_{\mathcal{D}}(\tilde{x}_{(k)}) + \tilde{u}_{(k)})$$

$$+ \eta'^2_{(k+1)}\partial g(x'_{(k+1)}) - \eta^2_{(k+1)}\partial g(x_{(k+1)})\|_2$$

$$\leqslant \frac{2C_{lip}}{n} + \frac{2C_{lip}\bar{\eta}}{n^2} + \frac{2\bar{\eta}^2 L_g}{n^2}.$$

From the formulations of $u_{(k+1)}$ and $u'_{(k+1)}$,

$$\|u_{(k+1)} - u'_{(k+1)}\|_2$$

$$=\|\eta_{(k+1)}(2\tilde{x}_{(k+1)} - \tilde{x}_{(k)}) - \eta'_{(k+1)}(2\tilde{x}'_{(k+1)} - \tilde{x}_{(k)})$$

$$+\eta'_{(k+1)}\mathrm{prox}_{\frac{1}{\eta'_{(k+1)}}\iota_{\mathcal{X}}}\left(\frac{1}{\eta'_{(k+1)}}\tilde{u}_{(k)} + (2\tilde{x}'_{(k+1)} - \tilde{x}_{(k)})\right)$$

$$-\eta_{(k+1)}\mathrm{prox}_{\frac{1}{\eta_{(k+1)}}\iota_{\mathcal{X}}}\left(\frac{1}{\eta_{(k+1)}}\tilde{u}_{(k)} + (2\tilde{x}_{(k+1)} - \tilde{x}_{(k)})\right)\|_2$$

$$\leqslant 2\|\eta_{(k+1)}\tilde{x}_{(k+1)} - \eta'_{(k+1)}\tilde{x}'_{(k+1)}\|_2 + \|(\eta_{(k+1)} - \eta'_{(k+1)})\tilde{x}_{(k)}\|_2 + 2\frac{\bar{\eta}}{n}D$$

$$\leqslant \frac{4C_{lip}}{n} + \frac{2C_{lip}\bar{\eta}}{n^2} + \frac{2\bar{\eta}^2 L_g}{n^2} + 2\frac{\bar{\eta}}{n}D,$$

which combing with (55) implies that

$$\|z_{(k+1)} - z'_{(k+1)}\|_2 \leqslant \|x_{(k+1)} - x'_{(k+1)}\|_2 + \|u_{(k+1)} - u'_{(k+1)}\|_2$$

$$=\frac{6C_{lip} + \bar{\eta}L_g + 2\bar{\eta}D + \frac{2C_{lip}\bar{\eta} + 2\bar{\eta}^2 L_g}{n}}{n}.$$

This completes the proof. $\qquad\square$

### A.8 PROOF OF THEOREM 2

*Proof.* From the sensitivity bound given in Lemma 1 and the update rule (22), for each $k \in \mathbb{N}$, a single update of $\tilde{z}_{(k)}$ leads to a privacy loss of $\varepsilon = \frac{\theta^2}{2n^2\delta^2}$. By using the composition property of RDP over the $K$ iterations, we can deduce that $\tilde{z}_{(K)}$ satisfies $\frac{K\theta^2}{2n^2\delta^2}$-CDP. Combining this with $\delta^2 = \frac{K\theta^2}{2n^2\rho}$ completes the proof. $\qquad\square$

### A.9 PROOF OF PROPOSITION 6

*Proof.* The guarantee for $\rho$-CDP can be directly obtained from Theorem 2.

Apply Proposition 5 with $\delta^2 = \frac{\theta^2}{2n^2\rho}K$, we know from (53) that

$$\mathbb{E}[\|\tilde{z}_{(K)} - \hat{z}_*\|_2^2 \mid \Gamma_0]$$

$$\leqslant \bar{\kappa}^{2K}\|\tilde{z}_{(0)} - \hat{z}_*\|_2^2 + \frac{1 - \bar{\kappa}^{2K}}{1 - \bar{\kappa}^2} \cdot \frac{\theta^2 dK}{n^2\rho}$$

$$\leqslant \bar{\kappa}^{2K}\|\tilde{z}_{(0)} - \hat{z}_*\|_2^2 + \frac{1}{1 - \bar{\kappa}^2} \cdot \frac{\theta^2 dK}{n^2\rho},$$

which indicates that

$$\mathbb{E}[\|\tilde{z}_{(K)} - \hat{z}_*\|_2 \mid \Gamma_0] \leqslant \sqrt{\mathbb{E}[\|\tilde{z}_{(K)} - \hat{z}_*\|_2^2 \mid \Gamma_0]}$$

$$\leqslant \bar{\kappa}^K\|\tilde{z}_{(0)} - \hat{z}_*\|_2 + \sqrt{\frac{1}{1 - \bar{\kappa}^2}} \cdot \frac{\theta\sqrt{dK}}{n\sqrt{\rho}}.$$

Let $K = \bar{\alpha} \log_{\bar{\kappa}} \left( \frac{1}{n\sqrt{\rho}} \right)$ with some $\bar{\alpha} \geqslant 1$. Let $\tilde{u}_{(0)} = -\nabla F_{\mathcal{D}}(\tilde{x}_{(0)}) - \partial g(\tilde{x}_{(0)})$. Then

$$\bar{\kappa}^K \|\tilde{z}_{(0)} - \hat{z}_*\|_2 \leqslant \frac{1}{(n\sqrt{\rho})^{\bar{\alpha}}} \|\tilde{z}_{(0)} - \hat{z}_*\|_2 \tag{56}$$

$$\leqslant \frac{1}{(n\sqrt{\rho})^{\bar{\alpha}}} \left( \|\tilde{x}_{(0)} - \hat{x}_*\|_2 + \|\tilde{u}_{(0)} - \hat{u}_*\|_2 \right) \tag{57}$$

$$\leqslant \frac{1}{(n\sqrt{\rho})^{\bar{\alpha}}} \left( 2D + \|\nabla F_{\mathcal{D}}(\tilde{x}_{(0)}) - \nabla F_{\mathcal{D}}(\hat{x}_*) + \partial g(\tilde{x}_{(0)}) - \partial g(\hat{x}_*)\|_2 \right) \tag{58}$$

$$\leqslant \frac{1}{(n\sqrt{\rho})^{\bar{\alpha}}} \left( 2D + L_{\nabla F_{\mathcal{D}}} \|\tilde{x}_{(0)} - \hat{x}_*\|_2 + 2L_g \right) \tag{59}$$

$$\leqslant \frac{1}{(n\sqrt{\rho})^{\bar{\alpha}}} \left( 2D + 2L_{\nabla F_{\mathcal{D}}} D + 2L_g \right). \tag{60}$$

Let $\tilde{D} := 2D + 2L_{\nabla F_{\mathcal{D}}} D + 2L_g$. Then

$$\mathbb{E}[\|\tilde{x}_{(K)} - \hat{x}_*\|_2]$$
$$\leqslant \mathbb{E}[\|\tilde{z}_{(K)} - \hat{z}_*\|_2]$$
$$\leqslant \frac{1}{(n\sqrt{\rho})^{\bar{\alpha}}} \tilde{D} + \frac{\theta}{n\sqrt{\rho}} \left[ \frac{d\bar{\alpha}}{1 - \bar{\kappa}^2} \cdot \log_{\bar{\kappa}} \left( \frac{1}{n\sqrt{\rho}} \right) \right]^{\frac{1}{2}}. \tag{61}$$

Furthermore, Assumption 1 implies that

$$\max_{x \in \mathcal{X}} \|\mathbb{E}_{s \sim \mathcal{P}} \left[ \nabla f(x; s) \right] \|_2 \leq \max_{x \in \mathcal{X}} \mathbb{E}_{s \sim \mathcal{P}} \|\nabla f(x; s)\|_2$$
$$\leq \mathbb{E}_{s \sim \mathcal{P}} \max_{x \in \mathcal{X}} \|\nabla f(x; s)\|_2$$
$$\leq G_1.$$

Then $F_{\mathcal{P}}$ defined in model (3) is $G_1$-Lipschitz. Together with Assumption 3, $\mathcal{F}$ is $(G_1 + L_g)$-Lipschitz continuous. Then

$$\mathbb{E} \left( |\mathcal{F}(\tilde{x}_{(K)}) - \mathcal{F}(\hat{x}_*)| \right) \leqslant \mathbb{E} \left( (G_1 + L_g) \|\tilde{x}_{(K)} - \hat{x}_*\|_2 \right)$$
$$= (G_1 + L_g) \mathbb{E} \left( \|\tilde{x}_{(K)} - \hat{x}_*\|_2 \right). \tag{62}$$

Therefore, the proof can be completed by substituting (61) into (62). $\qquad \square$

## A.10 PROOF OF PROPOSITION 7

*Proof.* Proposition 29 of (Lowy & Razaviyayn, 2023) implies that

$$\mathbb{E} \left[ \left( F_{\mathcal{P}}(\hat{x}_*) + g(\hat{x}_*) \right) - \left( F_{\mathcal{P}}(x^*) + g(x^*) \right) \right]$$
$$= \mathbb{E} \left[ \left( F_{\mathcal{D}}(\hat{x}_*) + g(\hat{x}_*) \right) - \left( F_{\mathcal{D}}(x^*) + g(x^*) \right) \right] + \mathbb{E} \left[ \left( F_{\mathcal{P}}(\hat{x}_*) + g(\hat{x}_*) \right) - \left( F_{\mathcal{D}}(\hat{x}_*) + g(\hat{x}_*) \right) \right]$$
$$\leq 0 + \frac{4G_2^2 + 2L_g^2}{\omega n} = \frac{4G_2^2 + 2L_g^2}{\omega n}.$$

The equality holds because $x^*$ is independent of $\mathcal{D}$, and the inequality holds because $\hat{x}_*$ is the empirical risk minimizer. $\qquad \square$

## A.11 PROOF OF THEOREM 3

*Proof.* By applying the triangle inequality to Proposition 6 and Proposition 7, we obtain that

$$\mathbb{E} \left( \mathcal{F}(\tilde{x}_{(K)}) - \mathcal{F}(x^*) \right)$$
$$\leqslant \mathbb{E} \left( |\mathcal{F}(\tilde{x}_{(K)}) - \mathcal{F}(\hat{x}_*)| \right) + \mathbb{E} \left( \mathcal{F}(\hat{x}_*) - \mathcal{F}(x^*) \right)$$
$$\leqslant (G_1 + L_g) \left( \frac{1}{(n\sqrt{\rho})^{\bar{\alpha}}} \tilde{D} + \frac{\theta}{n\sqrt{\rho}} \left[ \frac{d\bar{\alpha}}{1 - \bar{\kappa}^2} \cdot \log_{\bar{\kappa}} \left( \frac{1}{n\sqrt{\rho}} \right) \right]^{\frac{1}{2}} \right) + \frac{4G_2^2 + 2L_g^2}{\omega n}.$$

$\qquad \square$

### A.12 Linear Regression Tasks

The first evaluation is on an elastic-net regularized linear regression model (Zou & Hastie, 2005). For $x \in \mathbb{R}^d$ and $s := (\varphi, \upsilon) \sim \mathcal{P}$ with $\varphi \in \mathbb{R}^d$ and $\upsilon \in \mathbb{R}$, we define

$$f(x; s) := (\varphi^\top x - \upsilon)^2 + \frac{\omega}{2} \|x\|_2^2, \tag{63}$$

$$g(x) := \lambda \|x\|_1, \tag{64}$$

and

$$\mathcal{X} := \{x \in \mathbb{R}^d \mid \|x\|_2 \leqslant \varpi\}, \tag{65}$$

where $\omega, \lambda > 0$ are the regularization parameters and $\varpi > 0$ is a bound for the coefficient vector.

Given a data set $\mathcal{D} := \{s_1, s_2, \ldots, s_n\} \subseteq \mathcal{S}$ consisting of $n$ i.i.d. samples $s_i := (\varphi_i, \upsilon_i)$ drawn from a distribution $\mathcal{P}$, the empirical model can be formulated as

$$\min_{x \in \mathcal{X}} \{F_\mathcal{D}(x) + g(x)\} := \left\{ \frac{1}{n} \|\Phi x - F\|_2^2 + \frac{\omega}{2} \|x\|_2^2 + \lambda \|x\|_1 \right\}, \tag{66}$$

where $\Phi := [\varphi_1, \varphi_2, \ldots, \varphi_n]^\top$ and $F := [\upsilon_1, \upsilon_2, \ldots, \upsilon_n]^\top$.

It can be easily verified that $F_\mathcal{D}$ satisfies Assumptions 4 and 5 with $\omega > 0$ and $L_{\nabla F_\mathcal{D}} := \frac{2}{n} \|\Phi^\top \Phi\|_2 + \omega$, while the NDR $g$ satisfies Assumption 3 with $L_g = \lambda$. The following proposition derives a bound for $\mathbb{E}_{s \sim \mathcal{P}}[\sup_{x \in \mathcal{X}} \|\nabla f(x; s)\|_2]$.

**Proposition 8.** *For a given coefficient vector $x_* \in \mathbb{R}^d$, let the fidelity $f(\cdot; s)$ be defined by (63) with*

$$s := \begin{bmatrix} \varphi \\ \upsilon := \varphi^\top x_* + \varsigma \end{bmatrix} \sim \mathcal{P},$$ *where the random variables $\varphi \in \mathbb{R}^d$ and $\varsigma \in \mathbb{R}$ are independent of each other and satisfy that $\mathbb{E}_{s \sim \mathcal{P}}[\|\varphi\|_2^2] \leqslant d\varrho^2$ and $\mathbb{E}_{s \sim \mathcal{P}}[|\varsigma|] \leqslant \varkappa$ for some $\varrho, \varkappa \geqslant 0$. Then $\mathbb{E}_{s \sim \mathcal{P}}[\sup_{x \in \mathcal{X}} \|\nabla f(x; s)\|_2] \leqslant G_1$ with $G_1 := 2(\varpi + \|x_*\|_2)d\varrho^2 + 2\sqrt{d}\varrho\varkappa + \omega\varpi$.*

*Proof.* We can calculate that

$$\nabla f(x; s) = 2(\varphi^\top x - \upsilon)\varphi + \omega x.$$

Then

$$
\begin{aligned}
\|\nabla f(x; s)\|_2 &= \|2(\varphi^\top x - \upsilon)\varphi + \omega x\|_2 \\
&\leqslant 2\|(\varphi^\top x - \upsilon)\varphi\|_2 + \omega\|x\|_2 \\
&= 2\|\varphi\left(\varphi^\top(x - x_*) - \varsigma\right)\|_2 + \omega\|x\|_2 \\
&\leqslant 2|\varphi^\top(x - x_*)|\|\varphi\|_2 + 2|\varsigma|\|\varphi\|_2 + \omega\varpi \\
&\leqslant 2|\varphi^\top x|\|\varphi\|_2 + 2|\varphi^\top x_*|\|\varphi\|_2 + 2|\varsigma|\|\varphi\|_2 + \omega\varpi.
\end{aligned}
$$

If $\|\varphi\|_2 = 0$, we can directly know that $\|\nabla f(x; s)\|_2 \leqslant \omega\varpi$. If $\|\varphi\|_2 \neq 0$, then

$$
\begin{aligned}
\sup_{x \in \mathcal{X}} \|\nabla f(x; s)\|_2 &\leqslant \sup_{x \in \mathcal{X}} \left(2|\varphi^\top x|\|\varphi\|_2\right) + 2\|x_*\|_2\|\varphi\|_2^2 + 2|\varsigma|\|\varphi\|_2 + \omega\varpi \\
&= 2(\varpi + \|x_*\|_2)\|\varphi\|_2^2 + 2|\varsigma|\|\varphi\|_2 + \omega\varpi,
\end{aligned}
$$

which implies that

$$
\begin{aligned}
\mathbb{E}_{s \sim \mathcal{P}}[\sup_{x \in \mathcal{X}} \|\nabla f(x; s)\|_2] &\leqslant \mathbb{E}_{s \sim \mathcal{P}}[2(\varpi + \|x_*\|_2)\|\varphi\|_2^2 + 2|\varsigma|\|\varphi\|_2 + \omega\varpi] \\
&= 2\mathbb{E}_{s \sim \mathcal{P}}[(\varpi + \|x_*\|_2)\|\varphi\|_2^2] + 2\mathbb{E}_{s \sim \mathcal{P}}[|\varsigma|]\mathbb{E}_{s \sim \mathcal{P}}[\|\varphi\|_2] + \omega\varpi \\
&= 2(\varpi + \|x_*\|_2)d\varrho^2 + 2\sqrt{d}\varrho\varkappa + \omega\varpi.
\end{aligned}
$$

The last equality follows from the fact that $\mathbb{E}[\|\varphi\|_2] \leqslant \sqrt{\mathbb{E}[\|\varphi\|_2^2]} = \sqrt{d}\varrho$. $\qquad\square$

**Data generation.** For the synthetic data, the data dimensionalities are set to $n = 1000$ and $d = 64$. The true coefficient vector $x_*$ is defined as follows: $x_*^{(j)} = 0.5$ for $j \in [1:10]$, $x_*^{(j)} = 0.3$ for $j \in [11:20]$, $x_*^{(j)} = 0.2$ for $j \in [21:30]$ and 0 otherwise. For each $i \in [n]$, the sample $s_i$ is drawn from the distribution $\mathcal{P}$ such that $\varphi_i$ is generated independently from the standard Gaussian distribution $\mathcal{N}(0_{[d]}, I_{[d]})$, while the response variable $v_i$ is computed by $v_i = \varphi_i^\top x_* + \varsigma_i$, where $\varsigma_i \sim \mathcal{N}(0, 0.01)$.

For the real-world data from Kaggle, the initial data preparation entails normalizing the prices relative to their respective built year. The core task involves predicting this normalized house price using a collection of 16 predictor variables, such as the quantity of bathrooms and bedrooms. The experimental design specifies that houses built between [1900, 1950] constitute the training material, and those from the period (1950, 2000] are reserved for testing, which comprises 5,187 samples for training and 11,885 samples for testing.

We employ Algorithm 1 to privately solve the empirical model (66), which requires the explicit forms of $\text{prox}_{\eta_{(k)}g}$ and $\text{prox}_{\frac{1}{\eta_{(k)}}\iota_{\mathcal{X}}}$. It is well-known that for $g$ defined by (64),

$$\text{prox}_{\eta_{(k)}g}(x) := \text{sign}(x) * \max(|x| - \lambda\eta_{(k)}, 0_{[d]}), \tag{67}$$

where $*$ denotes the element-wise product. For the feasible set $\mathcal{X}$ defined by (65), we can deduce from the definition of the indicator function $\iota_{\mathcal{X}}$ in (10) that $\text{prox}_{\frac{1}{\eta_{(k)}}\iota_{\mathcal{X}}}$ is equivalent to the projection operator of $\mathcal{X}$:

$$\Pi_{\mathcal{X}}(x) := \begin{cases} x & \text{if } x \in \mathcal{X}, \\ \frac{\varpi}{\|x\|_2}x & \text{else.} \end{cases} \tag{68}$$

The detailed solving procedure for model (66) is provided in Algorithm 2.

---

**Algorithm 2:** NDR-HT for the empirical model (66)

**Require:** Data set $\mathcal{D} := \{s_1, s_2, \ldots, s_n\} \subseteq \mathcal{S}$, starting point $\tilde{z}_{(0)} \in \mathbb{R}^{2d}$. Set the upper bound of step size $\bar{\eta}$, the clipping parameter $C_{lip}$, the iteration count $K$, and the noise variance $\delta^2$.

1: **for** $k = 0, 1, 2, \ldots, K$ **do**
2:      Sample $\xi_{(k+1)}$ from the Gaussian distribution $\mathcal{N}(0_{[2d]}, \delta^2 I_{[2d]})$.
3:      Compute $\eta_{(k+1)} \leqslant \min\left(\frac{C_{lip}}{n\max(\|\frac{2}{n}\Phi^\top(\Phi\tilde{x}_{(k)}-F)+\omega\tilde{x}_{(k)}+\tilde{u}_{(k)}\|_2, \|\tilde{x}_{(k)}\|_2)}, \frac{\bar{\eta}}{n}\right)$.
4:      Compute $\hat{x}_{(k+1)} = \tilde{x}_{(k)} - \eta_{(k+1)}\left(\frac{2}{n}\Phi^\top(\Phi\tilde{x}_{(k)}-F)+\omega\tilde{x}_{(k)}+\tilde{u}_{(k)}\right)$.
5:      Compute $x_{(k+1)} = \text{sign}(\hat{x}_{(k+1)}) * \max(|\hat{x}_{(k+1)}| - \lambda\eta_{(k+1)}, 0_{[d]})$.
6:      Compute $\hat{u}_{(k+1)} = \frac{1}{\eta_{(k+1)}}\tilde{u}_{(k)} + (2\tilde{x}_{(k+1)} - \tilde{x}_{(k)})$.
7:      Compute $u_{(k+1)} = \eta_{(k+1)}\left(\hat{u}_{(k+1)} - \Pi_{\mathcal{X}}(\hat{u}_{(k+1)})\right)$.
8:      Compute $\tilde{z}_{(k+1)} = z_{(k+1)} + \xi_{(k+1)}$.
9: **end for**

**Ensure:** The solution $\tilde{x}_{(K)}$.

---

**Parameter setup.** The regularization parameters $\omega$ and $\lambda$ are both set to 0.1. The upper bound of step size $\bar{\eta}$ and the clipping parameter $C_{lip}$ are both set to 1. The parameter $\varpi$ for the feasible set is set to $\|x_*\|_2$. The privacy budget $\rho_{CDP}$ is evenly spread from 0.01 to 10 in the logarithmic scale. We set the variance $\delta^2$ of the noise vector and the number of iterations $K$ according to Theorem 3 with $\bar{\alpha} = 1$ to ensure the utility guarantee. For statistical reliability, we repeat the experiment for 10 times on each privacy budget.

## A.13   LOGISTIC REGRESSION TASK

We further validate our method using an elastic-net regularized logistic regression model. For $x \in \mathbb{R}^d$ and $s := (\varphi, v) \sim \mathcal{P}$ with $\varphi \in \mathbb{R}^d$ and $v \in \mathbb{R}$, we define the fidelity term $f(x; s)$, the regularization term $g(x)$ and the feasible set $\mathcal{X}$ as

$$f(x; s) := \log\left(1 + e^{-v \cdot \varphi^\top x}\right) + \frac{\omega}{2}\|x\|_2^2, \tag{69}$$

(64) and (65), respectively.

Given a data set $\mathcal{D} := \{s_1, s_2, \ldots, s_n\} \subseteq \mathcal{S}$ consisting of $n$ i.i.d. samples $s_i := (\varphi_i, \upsilon_i)$ drawn from a distribution $\mathcal{P}$, the empirical model can be formulated as

$$\min_{x \in \mathcal{X}} \{F_{\mathcal{D}}(x) + g(x)\} := \left\{ \frac{1}{n} \sum_{i=1}^{n} \log\left(1 + e^{-\upsilon_i \cdot \varphi_i^\top x}\right) + \frac{\omega}{2} \|x\|_2^2 + \lambda \|x\|_1 \right\}. \tag{70}$$

It can be easily verified that $F_{\mathcal{D}}$ satisfies Assumptions 4 and 5 with $\omega > 0$ and $L_{\nabla F_{\mathcal{D}}} := \frac{1}{n} \|(F * \Phi)^\top (F * \Phi)\|_2 + \omega$, where $\Phi := [\varphi_1, \varphi_2, \ldots, \varphi_n]^\top$ and $F := [\upsilon_1, \upsilon_2, \ldots, \upsilon_n]^\top$, while the NDR $g$ satisfies Assumption 3 with $L_g = \lambda$. The following proposition derives a bound for $\mathbb{E}_{s \sim \mathcal{P}}[\sup_{x \in \mathcal{X}} \|\nabla f(x; s)\|_2]$.

**Proposition 9.** *For a given coefficient vector $x_* \in \mathbb{R}^d$, let the fidelity $f(\cdot; s)$ be defined by (69) with*

$$s := \begin{bmatrix} \varphi \\ \upsilon := \varphi^\top x_* + \varsigma \end{bmatrix} \sim \mathcal{P}, \text{ where the random variables } \varphi \in \mathbb{R}^d \text{ and } \varsigma \in \mathbb{R} \text{ are independent}$$

*of each other and satisfy that $\mathbb{E}_{s \sim \mathcal{P}}[\|\varphi\|_2^2] \leqslant d\varrho^2$ and $\mathbb{E}_{s \sim \mathcal{P}}[|\varsigma|] \leqslant \varkappa$ for some $\varrho, \varkappa \geqslant 0$. Then $\mathbb{E}_{s \sim \mathcal{P}}[\sup_{x \in \mathcal{X}} \|\nabla f(x; s)\|_2] \leqslant G_1$ with $G_1 := \|x_*\|_2 d\varrho^2 + \sqrt{d}\varrho\varkappa + \omega\varpi$.*

*Proof.* We can calculate that
$$\nabla f(x; s) = -\frac{\upsilon}{1 + e^{\upsilon \cdot \varphi^\top x}} \varphi + \omega x.$$

Then
$$\|\nabla f(x; s)\|_2 \leqslant \left| \frac{\upsilon}{1 + e^{\upsilon \cdot \varphi^\top x}} \right| \|\varphi\|_2 + \omega \|x\|_2$$
$$\leqslant |\upsilon| \|\varphi\|_2 + \omega\varpi$$
$$= |\varphi^\top x_* + \varsigma| \|\varphi\|_2 + \omega\varpi$$
$$\leqslant \|\varphi\|_2^2 \|x_*\|_2 + |\varsigma| \|\varphi\|_2 + \omega\varpi,$$

which implies that
$$\mathbb{E}_{s \sim \mathcal{P}}[\sup_{x \in \mathcal{X}} \|\nabla f(x; s)\|_2] \leqslant \mathbb{E}_{s \sim \mathcal{P}}[\|\varphi\|_2^2 \|x_*\|_2 + |\varsigma| \|\varphi\|_2 + \omega\varpi]$$
$$= \|x_*\|_2 \mathbb{E}_{s \sim \mathcal{P}}[\|\varphi\|_2^2] + \mathbb{E}_{s \sim \mathcal{P}}[|\varsigma|] \mathbb{E}_{s \sim \mathcal{P}}[\|\varphi\|_2] + \omega\varpi$$
$$= \|x_*\|_2 d\varrho^2 + \sqrt{d}\varrho\varkappa + \omega\varpi.$$

The last equality follows from the fact that $\mathbb{E}[\|\varphi\|_2] \leqslant \sqrt{\mathbb{E}[\|\varphi\|_2^2]} = \sqrt{d}\varrho$. $\qquad\square$

Similar to the synthetic experiment, the solving algorithm for the logistic regression model is given by Algorithm 2, but replacing the computation of $\hat{x}_{(k+1)}$ in Step 4 by the following update:

$$\hat{x}_{(k+1)} = \tilde{x}_{(k)} - \eta_{(k+1)} \left( -\frac{1}{n} \Phi^\top * (F * \frac{1}{1 + e^{F * \Phi \tilde{x}_{(k)}}}) + \omega \tilde{x}_{(k)} + \tilde{u}_{(k)} \right).$$

**Data processing.** We conduct the experiments on the Adult data set from the UCI Machine Learning Repository. The data set comprises 48842 instances, each containing 14 personal attributes including age, sex, education level, and native country. The binary classification task predicts whether the income of an individual exceeds \$50,000. Following the preprocessing procedure described in (Huang et al., 2020), we transform categorical attributes into binary vectors, normalize all features, and map the original labels from $\{> 50000, \leq 50000\}$ to $\{+1, -1\}$. After preprocessing, each instance is represented by a 96-dimensional feature vector and a corresponding binary label. We randomly select 2,000 instances for training and another 2,000 instances for test, with both sets containing 50% positive and 50% negative samples.

**Parameter setup.** The regularization parameters $\omega$ and $\lambda$ are set to 0.1 and 0.001, respectively. The upper bound of step size $\bar{\eta}$ and the clipping parameter $C_{lip}$ are both set to 1. The parameter $\varpi$ for the feasible set is set to $\|(\Phi^\top \Phi + 0.01 I_d)^{-1} \Phi^\top F\|_2$, which provides a rough estimation of the coefficient vector norm. The privacy budget $\rho_{CDP}$ is evenly spread from 0.01 to 10 in the logarithmic scale. We set the variance $\delta^2$ of the noise vector and the number of iterations $K$ according to Theorem 3 with $\bar{\alpha} = 1$ to ensure the utility guarantee. For statistical reliability, we repeat the experiment for 10 times on each privacy budget.

### A.14 ADDITIONAL EXPERIMENTAL RESULTS

The experiments are carried out on a desktop workstation with an Intel Core i9-14900KF CPU, 64GB DDR5 6000-MHZ memory cards, and an Nvidia RTX-4080 graphics card with 16-GB independent memory.

**Convergence of NDR-HT.** We evaluate the convergence of NDR-HT under different privacy budgets using the optimality gap, defined by the difference between the objective value at each iteration and the optimal value. Results in Figure A1 show that the convergence performance is affected by very small privacy budgets (i.e., stronger privacy guarantees). This is because achieving a stricter privacy guarantee requires adding noise with a larger variance, which perturbs the optimization path.

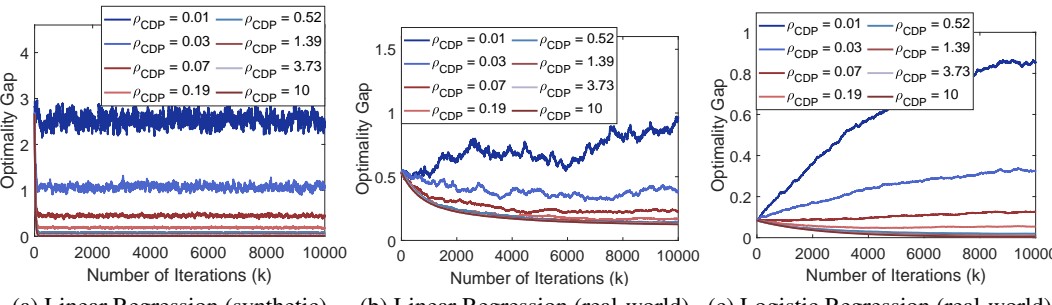

(a) Linear Regression (synthetic)  (b) Linear Regression (real-world)  (c) Logistic Regression (real-world)

Figure A1: Optimality gaps of NDR-HT during iterations under eight privacy budgets $\rho_{CDP}$.

In the experiment on synthetic data, the optimality gaps for strict privacy budgets ($\rho_{CDP} \leqslant 0.03$) decrease only marginally in the initial iterations, resulting in a considerable optimality gap. This result illustrates the inherent privacy-utility trade-off: stricter privacy requirements necessitate larger noise injection, which leads to slower convergence and larger optimality gaps. However, under moderate privacy budgets ($\rho_{CDP} > 0.03$), our method converges rapidly within a few iterations to a small optimality gap. On the other hand, the experiment on real-world data confirms the practicality of our method. For moderate privacy budgets ($\rho_{CDP} > 0.19$), our method successfully decreases the optimality gap and converges. These results show the effectiveness of NDR-HT on real-world data.

The optimality gaps of our method at the last iteration $K$ under eight privacy budgets $\rho_{CDP}$ are shown in Table A1. Our method achieves optimality with sufficiently large privacy budgets.

Table A1: Optimality gaps of NDR-HT at the last iteration under eight privacy budgets $\rho_{CDP}$.

| $\rho_{CDP}$ | 0.01 | 0.03 | 0.07 | 0.19 | 0.52 | 1.39 | 3.73 | 10 |
|---|---|---|---|---|---|---|---|---|
| Linear Regression (synthetic) | 2.4254 | 1.1365 | 0.4419 | 0.1779 | 0.0755 | 0.0349 | 0.0147 | 0.0063 |
| Linear Regression (real-world) | 0.9704 | 0.3910 | 0.2278 | 0.1653 | 0.1457 | 0.1321 | 0.1294 | 0.1262 |
| Logistic Regression (real-world) | 0.8558 | 0.3234 | 0.1259 | 0.0537 | 0.0198 | 0.0104 | 0.0048 | 0.0033 |

**Final objective function values.** The final objective function values ($\{F_{\mathcal{D}}(x) + g(x)\}$ in Eq. 9) of LNCSM, NCSGD and NDR-HT are shown in Table A2. NDR-HT achieves the best performance on both synthetic and real-world data across all privacy budgets considered.

**Accuracies for the classification task on the Adult data set.** Accuracies of LNCSM, NCSGD and our method for the classification task on the Adult data set are shown in Table A3. The model is trained on the training set and the classification accuracy is computed on the test set. Our method achieves the highest accuracies across all privacy budgets considered.

**MSEs for the house price prediction task on the Kaggle data set.** We compare the performance of NDR-HT with LNCSM and NCSGD using the MSEs of the prediction results. As shown in Table A4, NDR-HT outperforms both LNCSM and NCSGD when the privacy budget is moderate ($\rho_{CDP} \geqslant 0.07$).

Table A2: Final objective function values (mean $\pm$ STD) of LNCSM, NCSGD and NDR-HT.

| Linear Regression (synthetic) | | | | |
|---|---|---|---|---|
| $\rho_{CDP}$ | 0.01 | 0.03 | 0.07 | 0.19 |
| LNCSM | $5.5255 \pm 0.8959$ | $3.1990 \pm 0.3554$ | $2.7004 \pm 0.1698$ | $2.3064 \pm 0.1323$ |
| NCSGD | $5.3340 \pm 1.0688$ | $2.8808 \pm 0.3831$ | $2.2326 \pm 0.2844$ | $1.8458 \pm 0.2073$ |
| **NDR-HT** | $\mathbf{3.4928 \pm 0.3619}$ | $\mathbf{2.2039 \pm 0.2740}$ | $\mathbf{1.5093 \pm 0.0457}$ | $\mathbf{1.2453 \pm 0.0286}$ |
| $\rho_{CDP}$ | 0.52 | 1.39 | 3.73 | 10 |
| LNCSM | $2.1943 \pm 0.0764$ | $2.1311 \pm 0.0679$ | $2.0957 \pm 0.0486$ | $2.1166 \pm 0.0398$ |
| NCSGD | $1.7698 \pm 0.0890$ | $1.6535 \pm 0.0886$ | $1.6437 \pm 0.0407$ | $1.6456 \pm 0.0683$ |
| **NDR-HT** | $\mathbf{1.1429 \pm 0.0107}$ | $\mathbf{1.1023 \pm 0.0064}$ | $\mathbf{1.0821 \pm 0.0021}$ | $\mathbf{1.0737 \pm 0.0008}$ |
| Linear Regression (real-world) | | | | |
| $\rho_{CDP}$ | 0.01 | 0.03 | 0.07 | 0.19 |
| LNCSM | $1.6421 \pm 0.2130$ | $1.1242 \pm 0.1374$ | $0.8178 \pm 0.0635$ | $0.7517 \pm 0.0762$ |
| NCSGD | $1.7645 \pm 0.2777$ | $1.0729 \pm 0.1376$ | $0.8025 \pm 0.0732$ | $0.7305 \pm 0.0413$ |
| **NDR-HT** | $\mathbf{1.4700 \pm 0.0449}$ | $\mathbf{0.9376 \pm 0.0486}$ | $\mathbf{0.7401 \pm 0.0154}$ | $\mathbf{0.6679 \pm 0.0082}$ |
| $\rho_{CDP}$ | 0.52 | 1.39 | 3.73 | 10 |
| LNCSM | $0.6946 \pm 0.0314$ | $0.6896 \pm 0.0130$ | $0.6874 \pm 0.0087$ | $0.6844 \pm 0.0060$ |
| NCSGD | $0.7039 \pm 0.0243$ | $0.6748 \pm 0.0109$ | $0.6643 \pm 0.0146$ | $0.6718 \pm 0.0080$ |
| **NDR-HT** | $\mathbf{0.6340 \pm 0.0023}$ | $\mathbf{0.6246 \pm 0.0013}$ | $\mathbf{0.6190 \pm 0.0006}$ | $\mathbf{0.6175 \pm 0.0004}$ |
| Logistic Regression (real-world) | | | | |
| $\rho_{CDP}$ | 0.01 | 0.03 | 0.07 | 0.19 |
| LNCSM | $1.3297 \pm 0.3901$ | $1.0293 \pm 0.2060$ | $0.8274 \pm 0.1453$ | $0.7128 \pm 0.0749$ |
| NCSGD | $1.2737 \pm 0.3666$ | $0.9870 \pm 0.1724$ | $0.6700 \pm 0.0845$ | $0.5714 \pm 0.0383$ |
| **NDR-HT** | $\mathbf{1.2304 \pm 0.2944}$ | $\mathbf{0.6510 \pm 0.1044}$ | $\mathbf{0.4878 \pm 0.0503}$ | $\mathbf{0.4253 \pm 0.0179}$ |
| $\rho_{CDP}$ | 0.52 | 1.39 | 3.73 | 10 |
| LNCSM | $0.6207 \pm 0.0353$ | $0.5672 \pm 0.0171$ | $0.5473 \pm 0.0119$ | $0.5286 \pm 0.0067$ |
| NCSGD | $0.5568 \pm 0.0480$ | $0.5487 \pm 0.0308$ | $0.5517 \pm 0.0312$ | $0.5419 \pm 0.0351$ |
| **NDR-HT** | $\mathbf{0.4057 \pm 0.0128}$ | $\mathbf{0.3921 \pm 0.0089}$ | $\mathbf{0.3894 \pm 0.0061}$ | $\mathbf{0.3862 \pm 0.0036}$ |

Table A3: Accuracies (mean $\pm$ STD) of LNCSM, NCSGD and NDR-HT for classification task on the Adult data set.

| $\rho_{CDP}$ | 0.01 | 0.03 | 0.07 | 0.19 |
|---|---|---|---|---|
| LNCSM | $48.97 \pm 12.40\%$ | $49.18 \pm 14.79\%$ | $57.22 \pm 14.12\%$ | $54.08 \pm 19.43\%$ |
| NCSGD | $53.55 \pm 16.12\%$ | $51.90 \pm 14.42\%$ | $52.34 \pm 13.65\%$ | $54.51 \pm 15.50\%$ |
| **NDR-HT** | $\mathbf{72.68 \pm 2.67}\%$ | $\mathbf{74.41 \pm 0.90}\%$ | $\mathbf{74.91 \pm 0.16}\%$ | $\mathbf{75.01 \pm 0.02}\%$ |
| | 0.52 | 1.39 | 3.73 | 10 |
| LNCSM | $63.41 \pm 11.15\%$ | $64.33 \pm 9.31\%$ | $65.01 \pm 9.01\%$ | $69.33 \pm 4.86\%$ |
| NCSGD | $51.25 \pm 15.14\%$ | $59.88 \pm 7.72\%$ | $65.81 \pm 10.63\%$ | $67.63 \pm 9.42\%$ |
| **NDR-HT** | $\mathbf{75.00 \pm 0.00}\%$ | $\mathbf{75.00 \pm 0.00}\%$ | $\mathbf{75.00 \pm 0.00}\%$ | $\mathbf{75.00 \pm 0.00}\%$ |

Table A4: MSEs (mean $\pm$ STD) of LNCSM, NCSGD and NDR-HT for house price prediction task on the Kaggle data set.

| $\rho_{CDP}$ | 0.01 | 0.03 | 0.07 | 0.19 |
|---|---|---|---|---|
| LNCSM | $0.7623 \pm 0.3531$ | $0.6303 \pm 0.1776$ | $0.4745 \pm 0.0983$ | $0.3789 \pm 0.0966$ |
| NCSGD | $\mathbf{0.4764 \pm 0.0433}$ | $\mathbf{0.3608 \pm 0.0295}$ | $0.3229 \pm 0.0154$ | $0.3293 \pm 0.0048$ |
| **NDR-HT** | $0.6339 \pm 0.2082$ | $0.3760 \pm 0.0583$ | $\mathbf{0.3107 \pm 0.0239}$ | $\mathbf{0.2928 \pm 0.0160}$ |
| $\rho_{\mathrm{CDP}}$ | 0.52 | 1.39 | 3.73 | 10 |
| LNCSM | $0.3409 \pm 0.0224$ | $0.3073 \pm 0.0119$ | $0.2978 \pm 0.0238$ | $0.2963 \pm 0.0108$ |
| NCSGD | $0.3265 \pm 0.0073$ | $0.3314 \pm 0.0056$ | $0.3301 \pm 0.0038$ | $0.3306 \pm 0.0027$ |
| **NDR-HT** | $\mathbf{0.2814 \pm 0.0114}$ | $\mathbf{0.2778 \pm 0.0084}$ | $\mathbf{0.2748 \pm 0.0024}$ | $\mathbf{0.2760 \pm 0.0021}$ |

**Comparison of runtime for non-differentiable optimization methods:** We compare the runtimes of LNCSM and NDR-HT, both of which are applicable to NDR. Since NCSGD is a one-pass algorithm and cannot be applied to NDR, it is not comparable to the above two methods. For a fair comparison, the number of iterations for each method is set to ensure the optimal EPL guarantee. As shown in Table A5, In the linear regression experiment on the synthetic data, the average runtime of NDR-HT is approximately 0.2s, while that of LNCSM is around 6s. In the linear regression experiment on the real-world data, the average runtime of NDR-HT is approximately 1.3s, while that of LNCSM is around 105s. In the logistic regression experiment on the real-world data, NDR-HT has an average runtime of about 2s, compared with approximately 10s for LNCSM. These results verify the computational efficiency of NDR-HT, which are consistent with the theoretical results.

Table A5: Runtimes (mean $\pm$ STD) of LNCSM and NDR-HT.

| Linear Regression (synthetic) | | | |
|---|---|---|---|
| $\rho_{CDP}$ | 0.01 | 0.03 | 0.07 | 0.19 |
| LNCSM | $6.2653 \pm 1.8266$ | $5.9092 \pm 1.7053$ | $5.7177 \pm 1.8055$ | $6.3506 \pm 1.6455$ |
| **NDR-HT** | $\mathbf{0.1701 \pm 0.0318}$ | $\mathbf{0.1655 \pm 0.0204}$ | $\mathbf{0.1830 \pm 0.0253}$ | $\mathbf{0.2020 \pm 0.0282}$ |
| $\rho_{CDP}$ | 0.52 | 1.39 | 3.73 | 10 |
| LNCSM | $6.7026 \pm 1.4498$ | $6.4796 \pm 1.5693$ | $6.3739 \pm 1.5425$ | $6.5645 \pm 1.4003$ |
| **NDR-HT** | $\mathbf{0.2041 \pm 0.0277}$ | $\mathbf{0.2234 \pm 0.0121}$ | $\mathbf{0.2242 \pm 0.0299}$ | $\mathbf{0.2193 \pm 0.0233}$ |
| Linear Regression (real-world) | | | |
| $\rho_{CDP}$ | 0.01 | 0.03 | 0.07 | 0.19 |
| LNCSM | $103.8535 \pm 3.3164$ | $104.0350 \pm 1.4861$ | $102.7072 \pm 3.9724$ | $106.9256 \pm 6.4572$ |
| **NDR-HT** | $\mathbf{1.2946 \pm 0.3073}$ | $\mathbf{1.2742 \pm 0.2537}$ | $\mathbf{1.3922 \pm 0.0319}$ | $\mathbf{1.3326 \pm 0.1099}$ |
| $\rho_{CDP}$ | 0.52 | 1.39 | 3.73 | 10 |
| LNCSM | $106.8910 \pm 7.0970$ | $106.9350 \pm 9.0600$ | $103.9484 \pm 1.8854$ | $106.1226 \pm 7.0500$ |
| **NDR-HT** | $\mathbf{1.3579 \pm 0.0848}$ | $\mathbf{1.3319 \pm 0.1012}$ | $\mathbf{1.3482 \pm 0.0935}$ | $\mathbf{1.3601 \pm 0.0797}$ |
| Logistic Regression (real-world) | | | |
| $\rho_{CDP}$ | 0.01 | 0.03 | 0.07 | 0.19 |
| LNCSM | $10.0258 \pm 1.9050$ | $10.3085 \pm 1.9360$ | $9.3905 \pm 2.5529$ | $9.4244 \pm 2.7415$ |
| **NDR-HT** | $\mathbf{1.9978 \pm 0.6313}$ | $\mathbf{1.9786 \pm 0.6276}$ | $\mathbf{1.9851 \pm 0.6315}$ | $\mathbf{1.9835 \pm 0.6333}$ |
| $\rho_{CDP}$ | 0.52 | 1.39 | 3.73 | 10 |
| LNCSM | $9.9176 \pm 2.5442$ | $10.2384 \pm 1.8912$ | $10.4808 \pm 1.8914$ | $9.9437 \pm 1.8482$ |
| **NDR-HT** | $\mathbf{1.9891 \pm 0.5222}$ | $\mathbf{1.9878 \pm 0.6356}$ | $\mathbf{1.9925 \pm 0.5917}$ | $\mathbf{2.1265 \pm 0.4752}$ |

## A.15 SENSITIVITY ANALYSIS OF STRONG CONVEXITY PARAMETER $\omega$

We conduct a sensitivity analysis on the strong convexity parameter $\omega$, which plays a critical role in our theoretical bounds, to demonstrate the robustness of NDR-HT to parameter choice within a

Table A6: Sensitivity analysis of NDR-HT to the strong convexity parameter $\omega$. Objective function value is presented as mean $\pm$ standard deviation over 10 independent runs.

| | Logistic Regression (real-world) | | | | |
|---|---|---|---|---|---|
| $\rho_{CDP}$ | Strong Convexity Parameter $\omega$ | | | | |
| | 0.08 | 0.09 | 0.10 | 0.11 | 0.12 |
| 0.01 | $1.5015 \pm 0.0932$ | $1.4529 \pm 0.0907$ | $1.4303 \pm 0.0831$ | $1.4130 \pm 0.0721$ | $1.3475 \pm 0.0961$ |
| 0.03 | $0.9887 \pm 0.0478$ | $0.9498 \pm 0.0376$ | $0.9169 \pm 0.0441$ | $0.9456 \pm 0.0381$ | $0.8863 \pm 0.0345$ |
| 0.07 | $0.7666 \pm 0.0330$ | $0.7519 \pm 0.0167$ | $0.7474 \pm 0.0187$ | $0.7422 \pm 0.0166$ | $0.7395 \pm 0.0169$ |
| 0.19 | $0.6676 \pm 0.0090$ | $0.6655 \pm 0.0059$ | $0.6636 \pm 0.0071$ | $0.6690 \pm 0.0051$ | $0.6633 \pm 0.0048$ |
| 0.52 | $0.6304 \pm 0.0023$ | $0.6324 \pm 0.0017$ | $0.6362 \pm 0.0028$ | $0.6376 \pm 0.0026$ | $0.6394 \pm 0.0025$ |
| 1.39 | $0.6185 \pm 0.0016$ | $0.6209 \pm 0.0013$ | $0.6241 \pm 0.0014$ | $0.6268 \pm 0.0008$ | $0.6296 \pm 0.0011$ |
| 3.73 | $0.6126 \pm 0.0011$ | $0.6160 \pm 0.0005$ | $0.6191 \pm 0.0005$ | $0.6221 \pm 0.0007$ | $0.6251 \pm 0.0004$ |
| 10 | $0.6102 \pm 0.0005$ | $0.6139 \pm 0.0004$ | $0.6173 \pm 0.0005$ | $0.6206 \pm 0.0003$ | $0.6235 \pm 0.0002$ |

| | Linear Regression (synthetic) | | | | |
|---|---|---|---|---|---|
| $\rho_{CDP}$ | Strong Convexity Parameter $\omega$ | | | | |
| | 0.08 | 0.09 | 0.10 | 0.11 | 0.12 |
| 0.01 | $3.2880 \pm 0.6271$ | $2.2725 \pm 0.2983$ | $2.1807 \pm 0.4256$ | $1.7713 \pm 0.1992$ | $1.6199 \pm 0.4200$ |
| 0.03 | $1.3610 \pm 0.1873$ | $1.1249 \pm 0.1740$ | $0.9163 \pm 0.2062$ | $0.7741 \pm 0.1168$ | $0.6137 \pm 0.1148$ |
| 0.07 | $0.5599 \pm 0.0685$ | $0.4607 \pm 0.0999$ | $0.4161 \pm 0.0675$ | $0.3377 \pm 0.0546$ | $0.3199 \pm 0.0678$ |
| 0.19 | $0.2844 \pm 0.0550$ | $0.2267 \pm 0.0382$ | $0.2090 \pm 0.0355$ | $0.2203 \pm 0.0351$ | $0.2046 \pm 0.0436$ |
| 0.52 | $0.1795 \pm 0.0174$ | $0.1618 \pm 0.0212$ | $0.1734 \pm 0.0236$ | $0.1629 \pm 0.0276$ | $0.1646 \pm 0.0122$ |
| 1.39 | $0.1404 \pm 0.0118$ | $0.1336 \pm 0.0099$ | $0.1392 \pm 0.0131$ | $0.1512 \pm 0.0108$ | $0.1531 \pm 0.0123$ |
| 3.73 | $0.1267 \pm 0.0097$ | $0.1287 \pm 0.0084$ | $0.1312 \pm 0.0086$ | $0.1379 \pm 0.0088$ | $0.1394 \pm 0.0053$ |
| 10 | $0.1177 \pm 0.0046$ | $0.1226 \pm 0.0053$ | $0.1282 \pm 0.0046$ | $0.1307 \pm 0.0054$ | $0.1391 \pm 0.0035$ |

practical range. We test NDR-HT on both logistic and linear regression tasks by varying $\omega$ from 0.08 to 0.12.

The results in Table A6, Figure A2 and Figure A3 show the high stability of the final objective function value. For logistic regression on the real-world data set, the final objective value across the tested range is tightly clustered. Similarly, the final objective value for linear regression on the synthetic data set remains stable. In both cases, the standard deviations remain minimal, which confirms that the asymptotic performance and convergence of NDR-HT are highly robust to reasonable variations in the strong convexity parameter $\omega$. This validates the practical stability and reliability of NDR-HT.

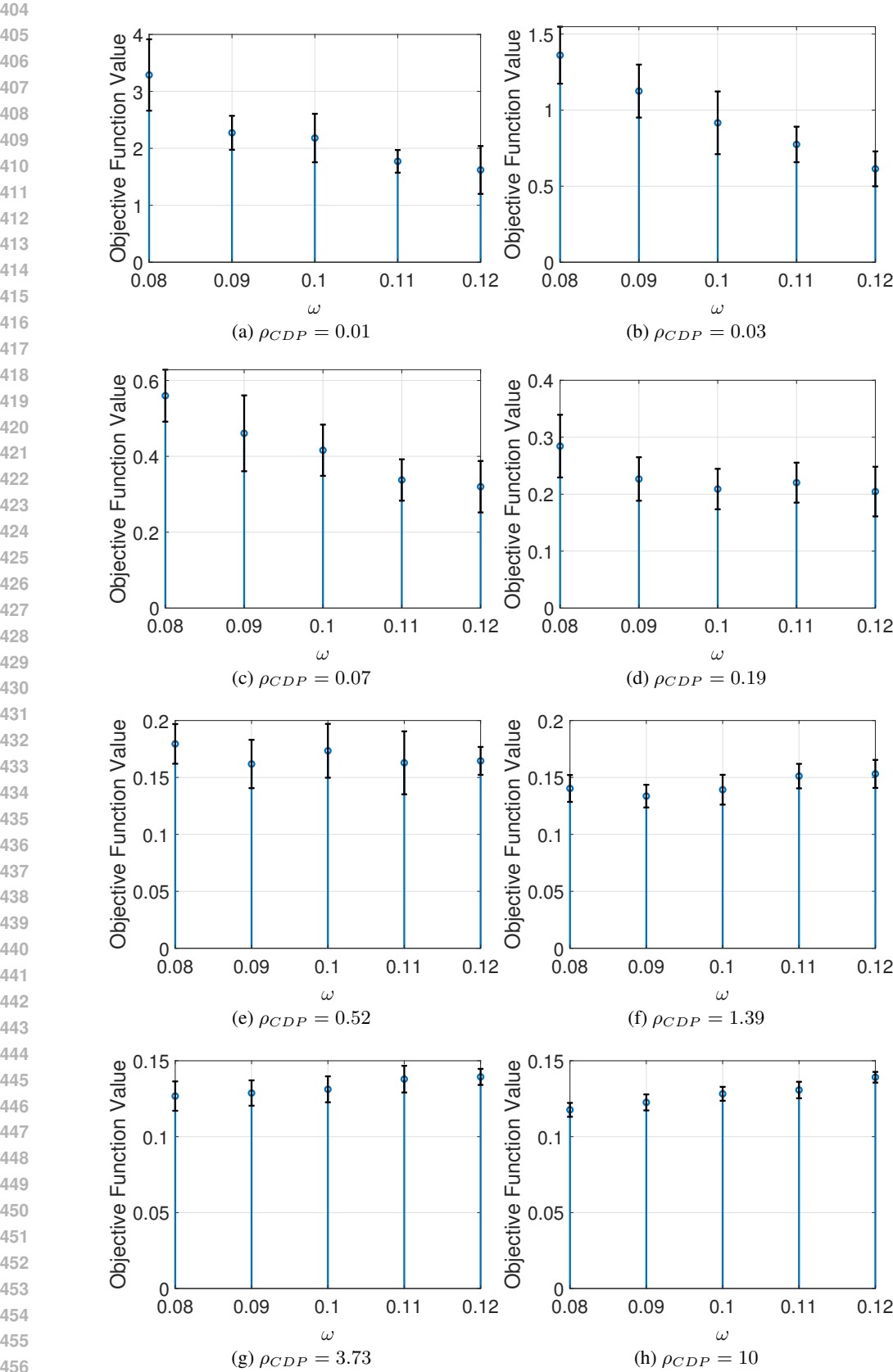

Figure A2: The objective function values of NDR-HT for $\omega$ ranging from 0.08 to 0.12 under eight privacy budgets ($\rho_{CDP}$) in the linear regression experiment on the synthetic data set. For the other experiments, $\omega$ is set to 0.1.

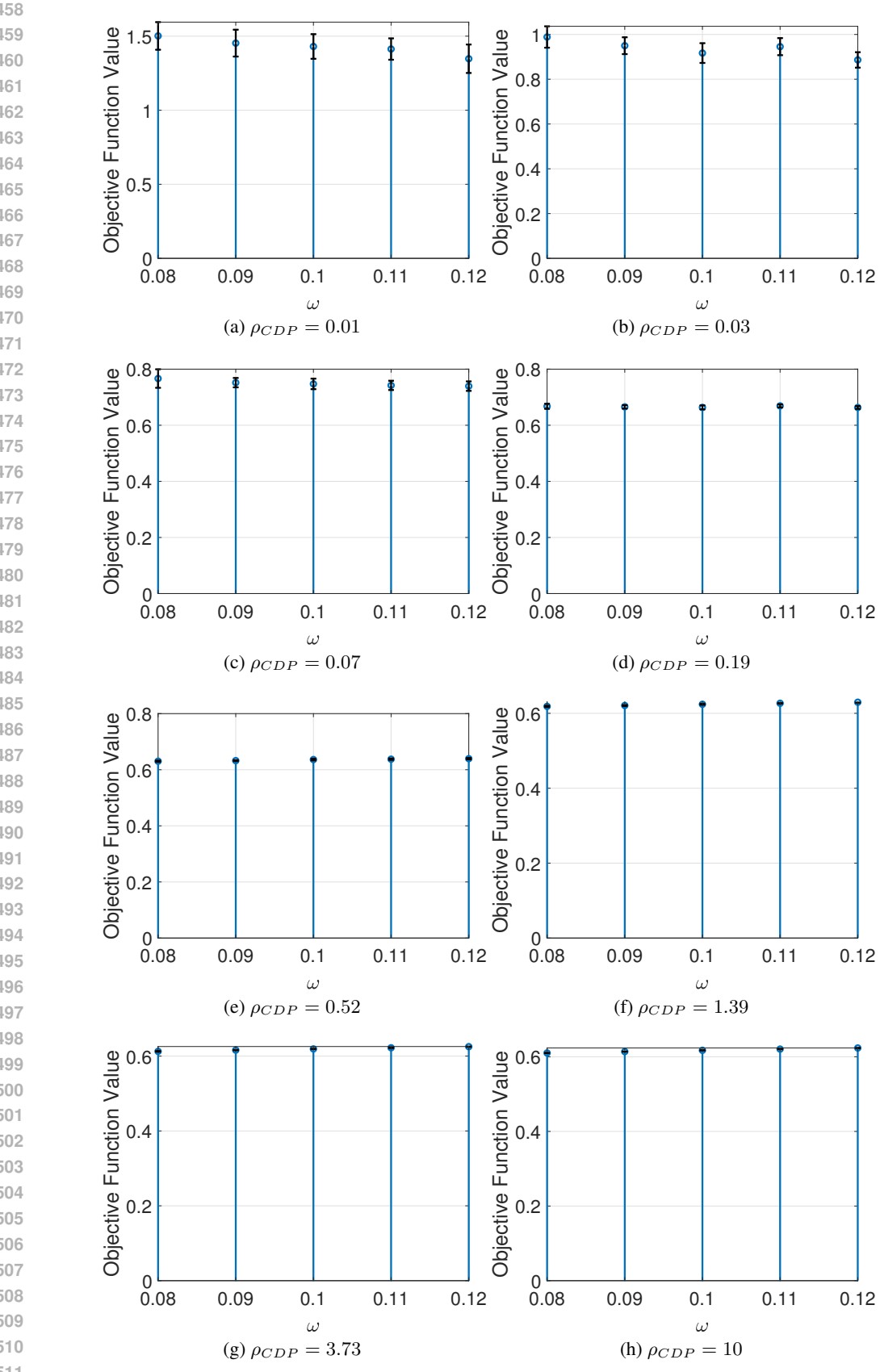

Figure A3: The objective function values of NDR-HT for $\omega$ ranging from 0.08 to 0.12 under eight privacy budgets ($\rho_{CDP}$) in the logistic regression experiment on the real-world data set. For the other experiments, $\omega$ is set to 0.1.