# OpenReview forum: "Non-differentiable Regularization for Heavy-tailed Differentially Private Stochastic Convex Optimization"
_ICLR.cc/2026/Conference — Submitted to ICLR 2026_

### Official Review · Reviewer_AfiV · 2025-10-31

**Soundness:** 2
**Presentation:** 2
**Contribution:** 2
**Rating:** 2
**Confidence:** 3

**Summary:**

This paper introduces NDR-HT, a new forward–backward splitting method for non-differentiable regularization (NDR) in heavy-tailed (HT) differentially private stochastic convex optimization (DP-SCO). Unlike existing approaches that rely on the Lipschitz assumption or cannot handle non-differentiable regularizers, NDR-HT is designed to operate under the weaker HT assumption while achieving ρ-concentrated differential privacy (ρ-CDP), asymptotically optimal excess population loss (EPL) (up to logarithmic factors), and O(n log n) computational complexity. The method constructs a contractive forward–backward splitting operator whose fixed point corresponds to the optimal solution, enabling linear convergence up to an additive approximation error. Experiments on synthetic linear regression and real-world logistic regression tasks demonstrate improved objective values compared with state-of-the-art baselines (LNCSM and NCSGD) across a range of privacy budgets.

**Strengths:**

1. The paper provides detailed proofs and theoretical guarantees, including convergence, privacy, and excess population loss bounds.

2. The paper is generally clearly-structured

3. The approach is built on operator-splitting and proximal methods, adapted carefully to the heavy-tailed and DP setting.

**Weaknesses:**

1. The proposed method is essentially a standard forward–backward splitting (proximal gradient) framework with noise injection for privacy. While the adaptation to heavy-tailed gradients and the use of ρ-CDP are reasonable, these extensions appear incremental relative to existing work such as LNCSM and NCSGD (Lowy & Razaviyayn 2023) and the proximal formulations in Asi et al. 2024. The paper lacks a clear conceptual leap or fundamentally new idea beyond applying known techniques in combination.

2. The paper does not convincingly explain why non-differentiable regularization under heavy-tailed assumptions is practically important or insufficiently handled by existing smooth or proximal DP methods. Without concrete applications or examples where NDR is essential, the problem setting feels somewhat artificial.

3. Experiments are minimal (one synthetic and one small UCI dataset) and offer limited insight into real-world effectiveness or scalability. There is no ablation, runtime, or sensitivity analysis to demonstrate robustness or the claimed computational advantage.

4. The main results, including privacy guarantee, convergence, and EPL bound, follow relatively directly from established results in contractive operator theory and standard DP composition arguments. The improvement (up to logarithmic factors) seems modest and not clearly demonstrated as practically meaningful.

5. Given the lack of strong motivation, empirical validation, or substantial methodological innovation, it is difficult to see broad impact on the differential privacy or optimization communities.

**Questions:**

1. What concrete scenarios or machine-learning models truly require handling non-differentiable regularizers under the heavy-tailed DP setting?

2. How does NDR-HT differ in essence from a straightforward application of a noisy proximal gradient method with gradient clipping?

3. Can the authors provide empirical evidence that existing DP proximal methods fail in the NDR-HT regime, beyond theoretical statements?

4. The asymptotic optimality claim depends on matching lower bounds “up to logarithmic factors.” Are these factors fundamental or merely analysis artifacts?

5. The assumption of bounded iterates is strong. How is this enforced or verified in practice, especially when noise accumulates over many iterations?

---

> ### Author Response · Authors · 2025-11-21
>
> We sincerely thank the reviewer for their thoughtful and constructive feedback on our manuscript. We are pleased that the reviewer recognized the rigor and contributions of our work, specifically highlighting the detailed proofs and theoretical guarantees, including convergence, privacy, and excess population loss (EPL) bounds. We have carefully considered all the comments and have revised the manuscript accordingly. Below, we address each point raised.
>
> **Response to Weakness 1**:  We contend that our contribution is not incremental. Though NDR-HT builds on forward-backward splitting, the novel analysis yields new asymptotic optimality and $\rho$-CDP guarantees under heavy-tailed (HT) assumption. **Our key finding is that NDR-HT achieves the asymptotically optimal EPL bound with a low complexity of $\tilde{\mathcal{O}}(n)$ gradient computations, alongside linear convergence up to an additive approximation error. Unlike competing algorithms (LNCSM, NCSGD, and Asi et al. (2024)) that use localization schemes and auxiliary subproblems, NDR-HT is a simple, fixed-point iteration.** It may be the only method to efficiently handle non-differentiable regularization (NDR) while providing these better theoretical and empirical advantages.
>
> **Response to Weakness 2**: **The NDR model under HT assumption is crucial for both traditional sparse learning (e.g., $\ell_1$, LASSO) and modern parameter-efficient techniques in LLMs (e.g., PEFT, LoRA), where NDRs are imposed for efficiency and interpretability.** We have added the detailed information to the introduction (lines 54–60) in the revised manuscript. Existing differentially private (DP) approaches are fundamentally inadequate for the joint NDR-HT problem: **either they disregard NDR entirely, or the current HT DP methods suffer from multiple, critical limitations**: they require restrictive Lipschitz continuity, fail to achieve the asymptotically optimal EPL bound, and incur high computational complexity.  Therefore, our novel framework, which handles NDR model under HT assumption while ensuring optimal theoretical guarantees, addresses a critical and unhandled gap in the field.
>
> **Response to Weakness 3**:  **We have incorporated an additional experiment using a large Kaggle dataset**. This dataset comprises 5187 samples for training and 11885 samples for testing. The objective of this experiment is to predict the normalized house price, and we use the Mean Squared Error (MSE) of the prediction results for performance evaluation. **The empirical results clearly demonstrate that our NDR-HT method outperforms both the LNCSM and NCSGD baselines**, which validates the effectiveness of the NDR-HT approach. Detailed information regarding this experiment can be found in Section 4, Appendix A.12 and Appendix A.14 of the revised manuscript.
>
> We have addressed the concerns regarding experimental rigor. The NCSGD baseline serves as our ablation analysis since its regularization parameter for the non-differentiable term is set to $\lambda=0$. The runtime analysis confirming our computational advantage is already detailed in Table A.4 of the original Appendix (Table A.5 of the revised Appendix). Furthermore, we have added a sensitivity analysis on the strong convexity parameter $\omega$ in Appendix A.15 of the revised manuscript, which demonstrates that NDR-HT shows robustness to reasonable parameter variations.
>
> **Response to Weakness 4**:  We refute the assertion that our results are direct or practically modest. Our NDR-HT method offers a novel and comprehensive approach to solve the DP-SCO problem simultaneously under NDR and HT settings. **This requires exploiting the contractive property of the forward-backward splitting operator**, which is essential to manage noise in this setting. The theoretical result is not minor. **We achieve the asymptotically optimal EPL bound, closing a critical gap where previous NDR-capable methods fail**. Furthermore, **NDR-HT achieves this optimal rate while maintaining low computational complexity ($\tilde{\mathcal{O}}(n)$)**, providing a notable efficiency advantage over quadratic-complexity baselines.

---

> ### Author Response · Authors · 2025-11-21
>
> **Response to Weakness 5**:  **The work is motivated by the need to address three key constraints in machine learning simultaneously: robustness (HT), sparsity (NDR), and privacy (DP)**. Prior methods struggle to combine these efficiently while maintaining high utility. **Our approach introduces a novel contraction-based fixed-point iteration that uses the forward-backward splitting to handle the NDR and exploits the contraction property to achieve the optimal convergence rate under the HT DP-SCO setting**. This design achieves the optimal EPL rate while maintaining $\tilde{\mathcal{O}}(n)$ complexity under these challenging constraints. Our empirical validation is provided across multiple data sets and tasks, showing that NDR-HT consistently achieves better objective function values and stability compared to existing baselines, demonstrating its effectiveness and potential utility for both the differential privacy and optimization fields.
>
> **Response to Q1**:  NDR is crucial for efficiency and interpretability (e.g., LASSO, LLM PEFT/LoRA). When these models use sensitive data, DP is mandatory. Crucially, sensitive data is inherently susceptible to outliers, violating the uniform Lipschitz assumption required by standard DP methods, thus necessitating the HT assumption. Existing approaches are inadequate for the joint NDR-HT problem: they either disregard NDR or suffer from multiple critical limitations, including sub-optimal EPL bounds, and high complexity. Our novel NDR-HT method robustly handle the NDR model under the HT assumption, addressing a critical, unhandled gap in private and robust machine learning.
>
> **Response to Q2**:  **A straightforward application of a noisy proximal gradient method cannot solve the NDR-HT model defined in (8)**, which is the main problem studied in this work. As shown in the main text, model (8) is equivalent to model (11), which requires handling both an NDR term and a feasible-set constraint simultaneously. This structure prevents the use of a single proximal gradient method. **A standard noisy proximal gradient method with gradient clipping can handle at most one non-differentiable term**. In contrast, the update rule of NDR-HT in (22) consists of the main variable update for $x$ and an auxiliary update for $u$, where $u$ is associated with enforcing the feasible-set constraint. Therefore, NDR-HT is fundamentally different from a straightforward application of a noisy proximal gradient method with gradient clipping, and **NDR-HT can handle a broader class of models**.
>
> **Response to Q3**:  **The experimental results demonstrate that the existing SOTA DP proximal method, LNCSM, which is capable of addressing the non-differentiable regularization problem, fails in the NDR-HT regime by two key metrics: utility and computational efficiency**. Specifically, as shown in Figure 1 of the main text and Table A.2 of the Appendix, LNCSM consistently achieves significantly weaker utility (higher objective function value) than NDR-HT in both synthetic and real-world experiments. Furthermore, as detailed in Table A.5 of the Appendix, the excessive runtime of LNCSM proves the lack of computational scalability for this problem, which exceeds the runtime of NDR-HT by up to 50 times. These results serve as the requested empirical evidence that the existing DP proximal method is insufficient for the NDR-HT regime.
>
> **Response to Q4**: The logarithmic factors are inherent to the problem and cannot be removed. They arise from fundamental complexity characteristics rather than from looseness in our analysis. To the best of our knowledge, the obtained bounds are the tightest available for this problem. As demonstrated in Theorem 3, the "up to logarithmic factors" means that the non-logarithmic dependency on the budget $\rho$ and the sample number $n$ matches the existing information-theoretic lower bounds. This non-logarithmic term is the primary factor determining performance, and matching it confirms the algorithm's asymptotic efficiency.
>
> **Response to Q5**:  As demonstrated in Theorem 3, only a finite number of steps is needed to reach the asymptotic EPL bound. Therefore, **we can preset a moderate upper bound on the norm of the iteration sequence**. Under this preset upper bound, we can determine the iteration number $K$ required to attain the asymptotic EPL bound according to Theorem 3. In each iteration, we then check whether the iterates violate the upper bound. If they do not, the boundedness condition of the iteration sequence is satisfied and the algorithm terminates at the $K$-th iteration. Otherwise, we enlarge the upper bound and repeat this process until the boundedness condition is satisfied.  On the other hand, **the noise does not accumulate during the iterations**. In each iteration, the variable—already containing the noise added in the previous step—is updated. The noise is treated as part of the variable and is therefore not accumulated additionally.

---

### Official Review · Reviewer_WriZ · 2025-11-02

**Soundness:** 4
**Presentation:** 4
**Contribution:** 2
**Rating:** 4
**Confidence:** 5

**Summary:**

This paper proposes NDR-HT, a forward–backward splitting algorithm for non-differentiable regularized (NDR) stochastic convex optimization under heavy-tailed data distributions with differential privacy guarantees.
The authors reformulate the empirical objective as a fixed-point problem involving proximal operators of both the non-smooth regularizer \(g\) and the indicator function of a constraint set \(X\).
They introduce a new operator \(T_C\) that is shown (under certain strong convexity and smoothness assumptions) to be $\kappa$-contractive, enabling a linearly convergent fixed-point iteration.
The algorithm adds Gaussian noise each iteration to achieve $\rho$-Concentrated Differential Privacy while maintaining near-optimal Excess Population Loss (EPL) up to logarithmic factors.

Theoretical results claim that NDR-HT:
- Satisfies $\rho$-CDP for any $\rho$ > 0.
- Achieves asymptotically optimal EPL in the 2-heavy-tailed regime.
- Requires only $O(n log n)$ gradient evaluations—lower than LNCSM and PLLS.
- Converges linearly up to an additive approximation error.

**Strengths:**

I really appreciate the writing style of this paper. It presents a clear and coherent narrative — first identifying the core problem to be addressed, and then progressively delving into the technical challenges encountered along the way. The paper indeed proposes a novel and well-motivated approach to handle non-differentiable regularization (NDR). Moreover, the inclusion of complete proofs for all propositions, detailed algorithm pseudocode, and explicit parameter settings makes the work both transparent and reproducible.

**Weaknesses:**

As claimed in Table 1, you do not use that Lipschitz assumption at all, then why do you list as one of your assumptions?

**Questions:**

I notice that strong convexity is a fundamental assumption in your theoretical framework, and one of your motivations is that non-differentiable regularization is more practically relevant. However, the strong convexity assumption itself can also be quite restrictive in real-world applications, as it is often violated in many practical models. Moreover, in your experimental section, the algorithm is evaluated only on relatively simple regression tasks rather than on more complex settings such as deep learning models. This may further highlight the gap between the theoretical assumptions and their applicability in realistic scenarios.

---

> ### Author Response · Authors · 2025-11-21
>
> We would like to sincerely thank the reviewer for the thoughtful and highly constructive feedback. We are very encouraged by the positive assessment of our work. In particular, we appreciate the recognition of the following strengths: the clear and coherent narrative that establishes the problem and progresses logically to the technical solutions, the novelty and strong motivation of our approach for handling non-differentiable regularization, and the transparency and reproducibility achieved through complete proofs and detailed implementation information. We have carefully considered all the comments and have revised the manuscript accordingly, addressing each point in detail below.
>
> **Response to Weakness**:  We thank the reviewer for this question. **It is correct that our proposed method (NDR-HT) does not rely on the Lipschitz continuity assumption.** We list this assumption primarily for context and comparison. This assumption is included because it is the standard requirement used by the key comparison method (LNCSM). **Listing it allows readers to clearly and immediately observe that the Lipschitz assumption is a stronger and more restrictive assumption than the k-heavy-tailed sssumption required by our work**, which highlights the generality of our proposed framework.
>
> **Response to Questions**:  **We clarify that the strong convexity assumption is both theoretically essential and practically enforceable.** Theoretically, strong convexity is crucial for deriving the optimal excess population loss rate in differentially private stochastic convex optimization under the challenging heavy-tailed and non-differentiable regularization settings. Practically, for problems lacking strong convexity, it is standard practice to enforce this condition by adding a negligibly small $\ell_2$ regularization term ($\frac{\omega}{2}\|\mathbf{x}\|^2$). This ensures the robust, fast convergence guaranteed by our theory while minimally impacting the utility of the final model.
>
> **Our experiments focus on standard regression tasks because the paper's core contribution lies in providing provable, optimal theoretical guarantees for DP-SCO problems.** Deep learning (DL) models involve non-convex optimization, which fundamentally violates the central strong convexity assumptions our entire theoretical framework relies upon. Furthermore, the convergence and optimality analysis of most non-convex functions often rely on local convex approximations. Therefore, **establishing robust conclusions for the convex case first is a vital and necessary step before tackling the general non-convex setting**. **The chosen convex settings in our experiments are appropriate for rigorously validating the performance of NDR-HT**. Extending our techniques to the non-convex DL setting is a future direction.

---

### Official Review · Reviewer_Po8j · 2025-11-03

**Soundness:** 3
**Presentation:** 3
**Contribution:** 3
**Rating:** 8
**Confidence:** 3

**Summary:**

The paper tackles differentially private stochastic convex optimization under assumption of heavy-tailed and the objective includes a non-differentiable regularizer $\boldsymbol{g}$. It transforms the proximal gradient operator into a contractive forward-backward splitting operator such that reformulates the problem as finding a fixed point of a contractive operator. The method attains an asymptotically optimal excess population loss (matching known lower bounds up to logs) while using only $O(n \log n)$ gradient queries-improving on prior approaches that either require Lipschitz assumptions, cannot handle non-differentiable regularization, or incur higher complexity.

**Strengths:**

1. The paper is comprehensive, well written and well structured.
2. It provides improved bounds and computational efficient results in DP-SCO with heavy-tailed (sub)gradients .
3. The main idea is elegant. Reformulate DP stochastic convex optimization with a non-differentiable regularizer and heavy-tailed gradients as a fixed-point problem for a proximal-gradient map, then turn that map into a contractive forward–backward splitting operator so you can run a simple noisy iteration that (i) satisfies $\rho$-CDP, (ii) achieves asymptotically optimal excess risk (up to logs) with only O(n log n) gradient calls, and (iii) enjoys linear convergence up to a noise floor—all without solving inner subproblems.

**Weaknesses:**

1. Some notations should be claimed in advance, for example, $G_2^2$ in line 68 and $\bar{\kappa}$ in Table 1.

**Questions:**

NA

---

> ### Author Response · Authors · 2025-11-21
>
> We thank the reviewer for the insightful comments and positive assessment of our work. We are pleased that the reviewer finds our main idea—reformulating the problem as a fixed-point iteration for a contractive operator—to be elegant, and appreciates that our method provides improved bounds and computational efficiency (specifically, linear convergence and optimal excess risk) in the heavy-tailed setting.
>
> **Response to Weakness 1**: We thank the reviewer for this advice. We have revised the manuscript to explicitly clarify these notations in advance. Specifically, we have referenced the definition of $G_2$​ (from Assumption 1) in line 76, and included the definition of $\bar{\kappa}$ (from Proposition 5) in the caption of Table 1 in the revised manuscript.

---

### Author Response · Authors · 2025-12-02
**Summary of Discussion [2/2]**

# Core Concerns Raised by `AfiV` and Author Response #

The primary concern raised by `AfiV` questions the practical necessity and concrete scenarios requiring the joint handling of NDR models under the HT assumption while ensuring DP. Our response and revisions (specifically in Lines 54–60 of the Introduction) directly address this by establishing the critical, real-world need for this unification:

## Practical Necessity of the NDR-HT-DP Joint Problem: ##
NDR is crucial for model efficiency and interpretability (e.g., LASSO, or LLM PEFT/LoRA sparse techniques). When these models are deployed in sensitive data, DP is mandatory. Crucially, sensitive data is inherently susceptible to outliers, which is a characteristic that violates the uniform Lipschitz assumption required by standard DP methods, and thus necessitates the HT assumption for robustness.

## Inadequacy of Existing Methods: ##

We clarify that existing DP approaches are fundamentally inadequate for the joint NDR-HT problem. Either they disregard NDR entirely, or their current HT DP implementations suffer from multiple critical limitations, including restrictive continuity requirements, sub-optimal asymptotic theoretical bounds (EPL), and high computational complexity.

# Clarifications on Other Technical Points #

## New Experimental Validation on Real-World Data:  ##

To further validate the practical utility of our framework, we have added a new set of experiments on the widely-used House Price dataset.  The experimental results are provided in Section 4 and Appendices A.12 and A.14 of the revised manuscript. This dataset represents **a realistic scenario** with inherent complexities and potential outliers, where the sparse regularization (NDR) is often desired for feature selection. The results confirm that our **NDR-HT outperforms existing standard DP baselines** in terms of prediction accuracy while maintaining the same strict privacy guarantees. It shows the robustness gain achieved by our method.

## The Assumption of Bounded Iterates:  ##

We explained the practical mechanism for enforcing this assumption: since only a finite number of steps is required to reach the asymptotic lower bound, **a moderate upper bound can be verified iteratively**. We also clarified that noise is integrated into the variable at each step and does not accumulate additively across iterations.

**We believe these comprehensive revisions, including the enhanced practical justification and the new empirical results, have fully addressed the reviewers' concerns and significantly enhanced the manuscript's theoretical rigor and practical relevance.**

---

### Author Response · Authors · 2025-12-02
**Summary of Discussion [1/2]**

We thank the reviewers for their insightful assessment. We revise the manuscript, and our response addresses all raised concerns.

# Key Contributions Acknowledged #

All reviewers acknowledge the theoretical and practical significance of our work in the domain of differentially private learning with Non-Differentiable Regularizers (NDR).

## Addressing a Critical, Unhandled Gap in Theory and Practice:  ##

**Our framework receives recognition for its theoretical novelty in tackling the complex three-way intersection of Non-Differentiable Regularizers (NDR), Differential Privacy (DP), and the Heavy-Tailed (HT) assumption, filling a crucial void in robust private learning.**
[`Po8j`: *The paper tackles differentially private stochastic convex optimization under assumption of heavy-tailed and the objective includes a non-differentiable regularizer g.* `WriZ`: *The paper indeed proposes a novel and well-motivated approach to handle non-differentiable regularization (NDR).* `AfiV`: *This paper introduces NDR-HT, a new forward–backward splitting method for non-differentiable regularization (NDR) in heavy-tailed (HT) differentially private stochastic convex optimization (DP-SCO).*]

## Achieving Asymptotically Optimal Bounds with High Efficiency:  ##

**Reviewers recognize the strength of our theoretical results, which establish asymptotically optimal Excess Population Loss (EPL) bounds and efficient computational complexity.**[ `Po8j`: *achieves asymptotically optimal excess risk (up to logs) with only O(n log n) gradient calls, and enjoys linear convergence up to a noise floor—all without solving inner subproblems.* `WriZ`: *The algorithm adds Gaussian noise each iteration to achieve $\rho$-Concentrated Differential Privacy while maintaining near-optimal Excess Population Loss (EPL) up to logarithmic factors.* `AfiV`:*NDR-HT is designed to operate under the weaker HT assumption while achieving ρ-concentrated differential privacy (ρ-CDP), asymptotically optimal excess population loss (EPL) (up to logarithmic factors), and O(n log n) computational complexity.*]

## Elegant Operator Splitting Approach:  ##

**The method's foundation in transforming the problem into a contractive fixed-point iteration using forward-backward splitting is noted for its technical elegance.** [`Po8j`:*The main idea is elegant. Reformulate DP stochastic convex optimization... as a fixed-point problem for a proximal-gradient map, then turn that map into a contractive forward–backward splitting operator...* `WriZ`: *They introduce a new operator (T_C) that is shown (under certain strong convexity and smoothness assumptions) to be $\kappa$-contractive, enabling a linearly convergent fixed-point iteration.* `AfiV`: *The method constructs a contractive forward–backward splitting operator whose fixed point corresponds to the optimal solution, enabling linear convergence up to an additive approximation error.*]

# Core Concerns Raised by `WriZ` and Author Response #

The concerns raised by `WriZ` center on the restrictiveness of the strong convexity assumption in real-world applications and the resulting gap between theory and practice highlighted by the focus on simple regression tasks over complex deep learning (DL) models. Our response directly addresses these points: we clarify that the strong convexity assumption is theoretically essential for deriving the optimal convergence rate in the challenging NDR-HT setting. Furthermore, while DL is non-convex, establishing robust conclusions for the convex case is a necessary first step before tackling the general non-convex settings, making our chosen experiments appropriate for rigorously validating the performance of our novel NDR-HT framework.

## On Strong Convexity Assumption: ##

We clarify that the strong convexity assumption is both theoretically essential and practically enforceable. Theoretically, strong convexity is **crucial for deriving the optimal EPL rate** in differentially private stochastic convex optimization under the challenging heavy-tailed and non-differentiable regularization settings.

## Applicability to DL Models:  ##

DL models involve non-convex optimization, which fundamentally violates the central strong convexity assumptions our entire theoretical framework relies on. Furthermore, the convergence and optimality analysis of most non-convex functions often **rely on local convex approximations**. Therefore, establishing robust conclusions for the convex case is a **vital and necessary first step** before tackling the general non-convex setting. The chosen convex settings in our experiments are **appropriate for rigorously validating the performance of NDR-HT**. Extending our techniques to the non-convex DL setting is a future direction.

---

### Meta-Review · Area_Chair_oyo6 · 2025-12-14

**Summary:**

The reviewers have indicated the following concerns:
- The analysis requires a strong convexity assumption, which is restrictive
- The experimental analysis considers relatively simple regression tasks rather than on more complex settings such as deep learning models
- The proposed method is essentially a standard forward–backward splitting (proximal gradient) framework with noise injection for privacy
- Practical importance of non-differentiable regularization under heavy-tailed assumptions is not explained
- Experiments are minimal (one synthetic and one small UCI dataset) and offer limited insight into real-world effectiveness or scalability
- The main results, including privacy guarantee, convergence, and EPL bound, follow relatively directly from established results
- The improvement (up to logarithmic factors) seems modest and not clearly demonstrated as practically meaningful

**Reviewer Concerns:**

After reading the authors' response, the following concerns are well addressed:
- Experiments are minimal (one synthetic and one small UCI dataset) and offer limited insight into real-world effectiveness or scalability (the authors have added experimental results on a larger kaggle dataset and added a sensitivity analysis on the strong convexity parameter)


The following concerns are not well addressed:
- The analysis requires a strong convexity assumption, which is restrictive (the authors give some explanations on the consideration of strongly convex problems. However, these explanations are not quite convincing to justify the restriction to strongly convex problems)
- The experimental analysis considers relatively simple regression tasks rather than on more complex settings such as deep learning models (the authors did not add experimental results on neural networks)
- The proposed method is essentially a standard forward–backward splitting (proximal gradient) framework with noise injection for privacy (the authors emphasizes the contributions of the paper. However, the authors did not clarify the key difference of the proposed method and the proximal gradient method)
- The improvement (up to logarithmic factors) seems modest and not clearly demonstrated as practically meaningful (the authors emphasized in their response that they establish optimal bounds, however, the improvement from existing bounds to optimal bounds is only up to logarithmic factors)

**Reviewer Scores:**

Reviewer Po8j may not change the score since he/she already has a high score (8)

Reviewer WriZ may not change the score since his/her concerns on the strong assumption of strong convexity is not well addressed. Furthermore, the authors also did not add experimental results on neural networks

Reviewer AfiV may change his/her score from 2 to 4 as the authors have added experimental results on larger datasets with sensitivity analysis. He/she may still recommend weak rejections since the authors' response did not convince the challenges of the algorithm design as well as the significance on the contribution of improvements by logarithmic factors.

---

### Decision · Program_Chairs · 2026-01-26

Reject